# Process of Introduction of Australian Braford Cattle to South America: Configuration of Population Structure and Genetic Diversity Evolution

**DOI:** 10.3390/ani12030275

**Published:** 2022-01-23

**Authors:** Araceli Rocío Marisel González, Francisco Javier Navas González, Gustavo Ángel Crudeli, Juan Vicente Delgado Bermejo, María Esperanza Camacho Vallejo, Celia Raquel Quirino

**Affiliations:** 1Laboratory of Animal Reproduction and Genetic Improvement, Northern Rio de Janeiro State University (Universidade Estadual do Norte Fluminense/UENF), Campos dos Goytacazes, Rio de Janeiro 28013-602, Brazil; araceligonzalez_18@hotmail.com (A.R.M.G.); crq@uenf.br (C.R.Q.); 2Institute of Agricultural Research and Training (IFAPA), Alameda del Obispo, 14014 Córdoba, Spain; mariae.camacho@juntadeandalucia.es; 3Department of Genetics, Faculty of Veterinary Sciences, University of Córdoba, 14071 Córdoba, Spain; juanviagr218@gmail.com; 4Theriogenology Area, Faculty of Veterinary Sciences, National University of the Chaco Austral (Universidad Nacional del Chaco Austral/UNCAus), Chaco 3700, Argentina; gacrudeli@hotmail.com

**Keywords:** Braford cattle, diversity evolution, inbreeding, Genetic Conservation Index, coancestry, nonrandom mating degree

## Abstract

**Simple Summary:**

The Braford breed originated in the USA and Australia from a cross between the Brahman and Hereford breeds to obtain animals suitable for the subtropical climate and resistant to hoof diseases, eye cancer, and ectoparasites, mainly ticks. This resistance to ticks was what attracted the attention of South American breeders, who acquired animals from Australia. The first breeder to do so was Uruguay around 1970. From then on, the breed was distributed across Argentina, Paraguay, and Brazil. Each country has its own association of breeders, and each one keeps the herdbook of the breed where the animals are registered. Selective breeding was conducted, thereby shaping genetic diversity over the years. The analysis of the pedigree database allowed us to evaluate these changes and the evolution of diversity over time. The objective of the present work was to analyze the population structure of the Braford breed in four countries, the repercussions of founders and ancestors, and the parameters of genetic diversity to suggest effective strategies for Braford breeders.

**Abstract:**

This study analyzes the evolution of the population structure and genetic diversity of Braford cattle in South America from 1949 to 2019 to suggest effective strategies for breeding in the future. The percentage of bulls historically increased. The average generational interval decreased to 11.78 years for the current population. Average inbreeding (F) and coancestry (C) are low and show a historically increasing trend (0.001% to 0.002%, respectively). The degree of nonrandom mating (α) increased from −0.0001 to 0.0001 denoting a change in the trend to mate similar individuals. The average relatedness coefficient (ΔR) increased in the current period from 0.002% to 0.004%. A single ancestor explained 4.55% to 7.22% of the population’s gene pool. While the effective population size based on the individual inbreeding rate (NeFi) was 462.963, when based on the individual coancestry rate (NeCi), it was 420.168. Genetic diversity loss is small and mainly ascribed to bottlenecks (0.12%) and to unequal contributions of the founders (0.02%). Even if adequate levels of diversity can be found, practices that consider the overuse of individual bulls (conditioned by nature or not), could lead to a long-term reduction in diversity. The present results permit tailoring genetic management strategies that are perfectly adapted to the needs that the population demands internationally.

## 1. Introduction

The most widespread historical records identify two initial Braford ‘lines’—Australian and American—whose origins date back to 1946 and 1947, respectively. The existence of these two ‘lines’ may stem from the fact that in the American Braford, Hereford bulls were used, while in the Australian Braford, Hereford cows participated in the cross.

These mating choices were aimed at meeting the different commercial interests for specific cattle meat quality characteristics produced in the particular environmental context of both countries. As a result, the additive genetic component of each diallel cross, with its base on the interaction between each parental sex and breed, led to the obtention of two differentiated products of crossbreeding (‘lines’) [1].

In this regard, breeding practices started with the use of different percentages of Brahman–Hereford crossbred bulls to find the best breed blood combinations seeking the balance between environmental adaptability (rusticity), longevity, maternal instinct (fertility, precocity, and docility), and meat performance and quality [2].

On the one hand, the origin of the American Braford ‘line’, is attributed to Alto “Bud” Lee Adams, Jr., who would begin mating a base herd of Brahman cows, primarily belonging to Partin and Hudgins breeding, to the Hereford bulls on his ranch in St. Lucie County, Florida in 1947.

However, despite the resulting steer and heifer calves characterized by excellent meat performance, the Hereford bulls required to produce those calves were extremely prone to the development of hoof, eye, and skin conditions and detrimental general performance and livability conditions [3,4]. Hence, the crossbreeding between Brahman and Hereford individuals occurred as a response to the Hereford bulls not being genetically suitable to their current environmental conditions, hence, not offering an economically viable alternative (i.e., slow growth rate, low productivity, high parasite load, and low resistance to extreme temperatures) (Figure 1).

Eventually, the identification of those Braford sires that produced calves that met environmental needs and the demands of the market became feasible and profitable. Those bulls and their descendants would form the foundation herd of the Braford breed in the United States [3,4].

On the other hand, parallel to the aforementioned (North) American ‘line’, registries set the origin of the Australian Braford ‘line’ at ‘Edengarry ranch’, north of Rockhampton in Queensland in 1946, when the Rea Brothers introduced Brahman bulls into their Hereford breeding females to help combat the effects of drought and ticks [5].

Although the origin of Queensland’s Braford cattle would be an attempt of Australian breeders to enhance cattle environmental adaptability via the improvement of biological performance in semi desert to tropical/subtropical climates, it would be the need of breeders in Southeastern Australia to reduce eye cancer incidence that would promote a second original Braford nucleus in New South Wales. However, Brahman cross cattle did not enjoy a good reputation for temperament or ease of handling, especially when run under extensive Australian conditions, hence it would take some time for Australian herdsmen to accept that the Brafords’ temperament was not a problem, and that they could breed an easily managed herd by selection [6].

It would take almost two decades, after the breed creation and development, before the Australian Braford Society [7] and the International Braford Association (IBA) were founded in 1962 and 1969, respectively, which would mark the moment when the registration of Braford animals would begin. However, another decade would pass until the first registered sale was made effective by the Adams ranch in Fort Pierce, Florida, on 14 December 1979.

As derived from heterosis, the combination of features of its parental breeds—Hereford and Brahman—makes Braford surpass the profitability of either of its parents, which is particularly relevant in challenging environmental contexts. For this reason, Brafords are especially suitable for commercial breeders seeking the advantages of their hybrid vigor, but also by their direct crossing with any breed. Braford is especially known for its early to medium maturity, that is, heifers reach puberty at a younger age, while steers on grass or in a feedlot finish quicker and with less food. This characteristic, as well as the breed’s exceptional feed conversion ability, results in great financial benefit for the feeder.

The Brahman inheritance is a phenotypically made patent through the presence of a hump, a low-set pizzle, a loose dewlap, and droopy ears. Additionally, Brahman cattle are known for their inherent resistance to eye cancer [8]. Thus, it is the Braford’s Brahman-inherited hooded eyes and good pigmentation around the eyes (blinkers or ‘anteojeras’), which provide them with resistance to eye cancer, pinkeye, and blight. Furthermore, Braford also inherited high tolerance to bloat from Brahman, which is of special relevance when cattle are grazed on clover and bloat-producing pastures [9], and it makes derived losses in Brafords infrequent.

On the Hereford side, the Braford breed has a smooth sleek coat of a basic red and white color although red color variations are acceptable, with markings that resemble those of a Hereford or Poll Hereford (‘pampa’ pattern). According to Silva [10], the preferable hair coat for tropical environments is characterized by light color, little thickness, high density (many hairs per unit area), and well settled thick hairs over a highly pigmented skin. The Braford’s inherent resistance to heat has been linked to relative reduced rectal temperatures (38.8 °C to 39.6 °C) on hot summer days, higher skin thickness (12.6 mm to 14.5 mm), and a higher ventilation capacity (54.8 and 63.6 breaths per minute) [11] compared to other breeds. As suggested by Bertipaglia et al. [12], this may be reinforced by the Braford’s lower coat thickness (3.73 cm), lower density coat (993.18 hairs/cm^2^), shorter hairs (10.41 mm), reduced hair diameter (30.98 μm), and an increased average sweating rate of 319.97 g·m^−2^·h^−1^. Furthermore, Braford coats tend to be thicker in winter in cool climates, which also protects them in extreme cold weather. In this regard, this can be considered as a favorable value to improve biological performance in animals bred in semidesert to tropical/subtropical climates.

Bearing these parental contributions in mind, the standard of excellence of Braford cattle states that the Brahman inheritance must be evident in the individual’s appearance, indicative of one fourth to three quarter Brahman characteristics, which are conferred to the Braford cross with sufficient environmental adaptability and resistance to undesirable conditions at the minimum meat performance cost, given that a fourth of the third-generation descendants derived from Brahman crosses (3/8 Brahman and 5/8 Hereford) present higher feed conversion ratios (higher dry matter intake to gain ratio [13]), which translated into poorer growth rates [14]. As a result, breeders are allowed to change the percentage of the Hereford–Brahman blood to fit to commercial demands [3]. In these regards, some markets limit the Brahman contribution to a maximum of 25% for some crosses (achieved by mating a Braford bull over Hereford, or mating a Poll Hereford bull over Braford cows), which in turn maximizes Braford cow traits. Desirable Braford growth rates are characterized by 18 to 20-month-old steers with a daily weight gain of 2.45 kg/day with a feed conversion ratio of 5.3:1.

In the context of the presumably parallel origin of the breed in North America and Australia and despite the apparently closer proximity of North and South America, it was Australian cattle which gave way to the South American population instead of the American population (although undocumented American contributions may have occurred along the course of the history of the breed).

The introduction of Australian bloodlines of the Braford breed into the South American continent would occur in the 1970s. As had occurred in Australia, the breed attracted breeders on the continent because one of the main animal health problems in semidesert to tropical/subtropical areas cattle production is the bovine tick, which causes decreased performance, hide devaluation, increased production costs with acaricide treatments, and transmission of infectious diseases [2,15].

The resistance of Brahmans to ticks started being studied more than five decades ago [16,17,18,19]. This resistance may be based on innate immune responses, structural genes, and genes that regulate the expression of tick skin hypersensibility with increased hyperreactions in nonresistant breeds [20,21], which translates into carcass and skin depreciation [22].

The foundation of new associations would quickly flourish across the South American continent with the creation of the Uruguayan Braford and Cebu Breeders Society in 1973, Argentinean Braford Association in 1984, Paraguayan Association of Braford Breeders, and the integration of the latter after the inclusion of Brazil through the formation of MERCOSUR Braford Federation in 1995. However, each breeder association implements its specific breeding criteria based on the specific requirements of the market, which in turn conditions basic requirements for studbook inclusion, which are necessary for individuals to be officially registered within the Braford population [3,4], which promotes the differential evolution of diversity across worldwide regions.

Since the origin of Braford cattle, breeders implemented selective breeding and a broad range of mating plans among reproductively active individuals within the context of every international breeding program. However, this unavoidably and differentially altered the genetic structure of the population in each particular place, which translated into an unequal response of genetic diversity parameters (i.e., increase in generation intervals and inbreeding levels) [23]. The analysis of the information present in breed pedigrees permits the tracing of the genetic variability and its evolution across generations. Furthermore, the evaluation of ancestral contributions may permit isolating the repercussions of historical animal lines or families, evaluating their long-lasting effects in the population and the diversity gain or loss that maintaining such animals or their descendants may imply as a manner to enhance genetic gain for economically important traits, and preventing inbreeding depression derived from potentially deleterious effects [24].

Studying the internationalization of breeds from the place where these originated and the processes that they go through once in their new locations, becomes an invaluable critical tool to understand the genetic diversity status of breeds at present. The information that is present in pedigrees enables the assessment of the genetic and demographical structure of animal populations and can be used to trace such aforementioned process of internationalization. Likewise, this knowledge helps us to infer the trends that breeds will describe in the future.

Given that diversity studies are rather frequently implemented in endangered and limited populations, the computation of genetic diversity parameters in large populations can be challenging as the software available is not prepared for it [25]. However, a genetic diversity status, population structure evaluation, and breeding practice assessment are compulsory steps for the correct implementation of breeding programs. Contextually, even if population survival may be ensured by a large number of individuals, incorrect breeding policies may lead to a decrease in the genetic health of the population, an effective population size reduction, and a consequent genetic diversity loss. This in turn may translate into a decreased adaptability to the diverse environmental conditions found worldwide, reduced performance, and economic losses that in the end directly affect the sustainability of breeds, which no longer render them profitable.

Therefore, the aim of this study is to perform the analysis of the pedigree completeness downwards from current individuals in the population of the South American Braford to its ancestors. The present study checks the repercussions of ancestors and founders in the South American Braford population since its origin in the Australian Braford ‘line’, evaluating the current and historical structure of the population, its genetic diversity and connections between genetic and demographic parameters, measuring the existing gene flow and quantifying the risk of genetic diversity loss to suggest effective breeding strategies. The present study may serve as a model for the study of diversity and population structures for breeds in geographically diverse environmental frameworks [26].

## 2. Materials and Methods

### 2.1. Pedigree Database and Software Tool

The pedigree dataset for the herdbook of the Braford cattle breed was supplied by the Argentinean Association of Braford (Asociación Braford Argentina). The herdbook of the Braford cattle breed used in this study comprised the Brafords which were historically and currently registered across Argentina, Uruguay, and Paraguay and their foundation basis in Australia. Appendix A provides further official information in regard to the territorial distribution, husbandry practices, reproductive management, and conservation strategies currently implemented among other relevant information about the Braford cattle breed. First, the complete pedigree dataset used in this study consisted of the historical population of the breed. Historical population dataset comprised 358,041 dead and alive animals (124,713 bulls and 233,328 cows, born between September 1949 and December 2019) (Table 1). Apart from performing demographic and genetic analyses on the complete pedigree dataset (historical population), all analyses were performed on a second smaller set (current population) comprising all alive animals in the population of the breed (115,757 animals, 53,770 bulls, and 61,987 cows, born from September 1998 to December 2019).

The calculations linked to genetic diversity, probabilities of gene origin, and founder analyses can only be performed either solely considering animals with both parents known or by comparing this set of animals to the historical and current datasets as suggested by Arias et al. [27] and Alanzor Puente et al. [28]. As a result, a third dataset was considered (reference population). The reference population set comprised 16,704 animals from the current population (12,810 bulls and 3894 cows), from which all sires and dams were known (1st generation was completely known). Navas et al. [29] suggested that considering population sets for which the first generation of animals is known may offer an opportunity to determine the distortion occurring in diversity parameters typical for unbalanced genealogies, in which information on one of the sexes has been historically considered more relevant (Table 2).

Further information on the composition of samples can be found in Table 1 and Table 2. ENDOG (v4.9) software [30] was used to perform demographic and genetic analyses to quantify and trace pedigree diversity back to ancestors and founders. From a theoretical perspective, ancestors with no known parent were considered as founders (generation 0) [30].

### 2.2. Population Summary Statistics

The number of births was computed to determine the maximum and mean number of offspring per bull and cow. Pedigree Completeness Index (PCI) of each population dataset was computed following the premises in Navas et al. [28]. Average age of parents (at birth of offspring) and generation interval [31] were calculated for the 4 gametic pathways: bull and cow to son and daughter, respectively using the birth date record for each animal together with those of its parents’. A summary of statistics derived from pedigree analyses is shown in Table 1. Total and mated cow to bull ratios were also calculated dividing total ewes by total bulls and breeding cows by breeding bulls, respectively. Summary of the analysis of the maximum number of traced generations, pedigree completeness (for 1st, 2nd, 3rd, 4th, and 5th generations of ancestors), number of maximum generations, number of complete generations, and number of equivalent generations in the three studied population datasets (historical and current datasets) are shown Table 2. Generation intervals (years) and the mean age (years) of the parents at the birth of their offspring selected for breeding for the four gametic routes in the Braford cattle breed are shown in Appendix A.

### 2.3. Inbreeding, Coancestry, and Assortative Mating Degree

The coefficient of coancestry (*C*) (equivalent to kinship) between pair of individuals is the probability that genes, taken at random from each of the concerned individuals, are identical by descent (IBD) [32]. The chance that both homologous genes in the same zygote are IBD is called inbreeding (*F*) (or coefficient of inbreeding). As a result, the *C* between two individuals is the *F* of their potential offspring. Individual *F* was computed using the methods described in Meuwissen and Luo [33]. Each individual’s average coefficient of relatedness (ΔR) refers to the probability that two related individuals have inherited a particular allele of the single locus/gene from their common ancestor (such an allele is referred to as IBD) and was calculated according to Gutiérrez et al. [30]. According to Leroy et al. [34], although *F* and *C* are both IBD estimators, the values for these probabilities may differ depending on whether the alleles considered belong to a single individual or two individuals, respectively. The individual rate of inbreeding ($\bar{\Delta F}$) for each generation was calculated according to Gutiérrez et al. [35] as follows: $\Delta F_{b}=1-\sqrt[{t_{b}-1}]{1-F_{b}}$, where *t_b_* is the number of complete equivalent generations, and *F_b_* is the *F* of the individual *b*.

Average *F* per generation was used to test the explanatory and predictive performance of a linear and a quadratic regression function extending *F* fifteen generations onwards.

The individual rate of coancestry ($\bar{\Delta C}$) for each generation was computed following the methods described in Cervantes et al. [36]: $C_{ba}=1-\sqrt[{\frac{tb+ta}{2}}]{1-C_{ba}}$, where *t_b_* and *t_a_* are the number of equivalent complete generations, and *C_ba_* is the *C* for the individuals *b* and *a*. Homogamy, assortative or nonrandom mating degree (α) describes the mating pattern or form of sexual selection in which individuals with similar phenotypes or genotypes mate with one another more frequently than would be expected under a heterogamy, random, or disassortative mating pattern (individuals with a different genotype or phenotype are more likely to mate with one another than would be expected at random). Assortative mating is less frequent than disassortative mating in animals. Nonrandom mating was computed following the methods in Caballero and Toro [37]: (1−F)=(1−C)(1−α).

### 2.4. Ancestral Contributions and Probabilities of Gene Origin

The effective number of founders (*f_e_*) was calculated using the following formula: $f_{e}=\frac{1}{\sum_{k=1}^{f}q_{k}^{2}}$, where *q_k_* is the probability of gene origin of the *k*^th^ founder, and *f* is the real number of founders [38]. As explained, using ENDOG (v4.9) software [30], *f_e_* was computed using the AR coefficients of founder individuals; hence it would be homologous to that computed following the premises in Lacy [38] if the reference population used was the whole pedigree. The effective number of ancestors (*f_a_*) was determined as follows: $f_{a}=\frac{1}{\sum_{k=1}^{f}p_{k}^{2}}$, where *p_k_* is the marginal contribution of a *k*^th^ ancestor [39]. *f_a_* is the minimum number of ancestors—which can be founders or not—needed to explain the genetic diversity of the current population. If there were no population bottlenecks, *fa* would equal *fe*, and the number and severity of bottlenecks would be reflected in the difference between *fa* and *fe*. The effective number of founder genomes (*f_g_*) was computed as the inverse of twice the average *C* as reported in Caballero and Toro [37] and can be defined as the number of founders that would be expected to produce the same genetic diversity as in the population under study if the founders were equally represented and no loss of alleles occurred. ENDOG (v4.9) software [30] took those animals in the pedigree with both parents known as the reference population to compute *f_g_* as the default. The expected marginal contribution of each major ancestor *j* was computed according to Boichard et al. [39], and the contributions to inbreeding of nodal common ancestors (inbreeding loops) were computed according to Colleau and Sargolzaei [40]. CFC version 1.0 software was used to compute ancestral contributions and probabilities of gene origin [41].

The effective population size (N_e_) is the size of an ideal population (which meets all the Hardy–Weinberg assumptions: (1) random mating (i.e., population structure is absent, and matings occur in proportion to genotype frequencies), (2) the absence of natural selection, (3) a very large population size (i.e., genetic drift or random fluctuations in the frequencies of alleles from generation to generation due to chance events is negligible), (4) no gene flow or migration, (5) no mutation, and (6) the locus is autosomal) that would lose heterozygosity or the proportion of heterozygotes in the population (i.e., individuals with two different alleles at a locus) at a rate equal to that of the observed population. The mean effective population size ($\bar{N_{e}}$) [42] was calculated as $\bar{N_{e}}=\frac{1}{2\bar{\Delta IBD}}$. The number of equivalent subpopulations was computed according to Cervantes et al. [43] using $S=\frac{\bar{N_{e}C_{i}}}{N_{e}F_{i}}$, where $\bar{N_{e}C_{i}}=\frac{1}{\left(2\bar{\Delta C}\right)}$ is the mean effective population size computed considering *C*, and $\bar{N_{e}F_{i}}=\frac{1}{\left(2\bar{\Delta F}\right)}$ is the mean effective population size computed considering *F*. Genetic diversity (*GD*) was calculated using $GD=1-\frac{1}{2f_{g}}$. GD lost in the population since the founder generation was estimated using 1−GD. GD loss derived from the unequal contribution of founders was estimated according to Caballero and Toro [37] using $1-GD^{*}$, where $GD^{*}=1-\frac{1}{2f_{e}}$. The unequal contribution of founders relates to the fact that genetic contributions from founders of specific populations can be of different proportions due to past directional mating (human-mediated or not) during the process configuration of the population. The difference between *GD* and *GD** indicates the *GD* loss owed to genetic drift accumulated since the foundation of the population [38], and the effective number of nonfounders (*N_ef_*) was computed using $N_{ef}=\frac{1}{\frac{1}{f_{ge}}-\frac{1}{f_{e}}}$ considering the formula by Caballero and Toro [37].

### 2.5. Country, Province, and Herd Relationships

The minimum Nei’s genetic distance [44] between subpopulations *i* and *j* was computed as in Navas et al. [28] to assess country, province, and herd relationships. Nei’s genetic distance is a measure of the genetic divergence over time from common ancestor. Dendrograms for country, province, and herd relationships for the Braford breed were constructed using the DendroUPGMA application found in Garcia-Vallvé and Puigbo [45] and the construct Unweighted Pair-Group Method using Arithmetic averages (UPGMA) Tree task from the Phylogeny procedure of MEGA X 10.0.5 [46].

Wright’s F-statistics or fixation indexes describe the statistically expected degree of a reduction in heterozygosity when compared to the Hardy–Weinberg Equilibrium (HWE) expectations. This set of parameters measures the genetic structure of a population. Wright’s F-statistics were computed following the premises in Caballero and Toro [37]. F-statistics are as follows: *F*_IS_ (inbreeding coefficient (*F*) relative to the subpopulation or the proportion of the variance in the subpopulation contained in an individual); F_ST_ (correlation between random gametes drawn from the subpopulation relative to the total population or the proportion of the total genetic variance contained in a subpopulation (the _S_ subscript) relative to the total genetic variance (the _T_ subscript); and *F*_IT_ (*F* relative to the total population). *F*_IT_ can be partitioned into F_ST_ due to the Wahlund effect (reduction of heterozygosity in a population caused by subpopulation structure) and *F*_IS_ due to inbreeding. Additionally, self-coancestry (f) was computed as the probability that two alleles taken at random from an individual (independently and with replacement) are IBD. As the IBD condition can result from sampling the same allele twice or sampling two alleles that happen to be IBD, the coancestry of an individual with itself f(A/A) equals $\frac{1+F\left(A\right)}{2}$, where *F*(*A*) is *F* for that particular individual. This must be understood considering coancestry in one generation becomes inbreeding in the following one.

Contextually, selfing or autocoancestry [47] would imply a case in which a certain individual mates itself (hermaphroditism). In the context of domestic animal populations, the possibility of autofecundation is almost always discarded and considered 0, as it equals *F* when autofecundation occurs, which logically did not happen in our case.

## 3. Results

### 3.1. Census Evolution, Herd Number, Generation Intervals, and Bull to Cow Ratio

The average number of herds decreased from 223 historical herds to the 180 currently existing herds. These herds went from being present in 22 to 16 provinces across the four countries that were evaluated (Australia, Argentina, Paraguay, and Uruguay). Table 1 shows a summary of statistics derived from pedigree analysis in the historical (*n* = 358,041) and current (*n* = 115,757) populations. The average herd size also decreased from 1605.57 to 643.10 animals per herd in the same population datasets, respectively. While the total percentage of bulls and cows historically constituted 34.83% and 65.17%, respectively, these percentages increased in bulls to 46.45% and decreased in cows to 53.55% in the current population, respectively. Appendix A presents the descriptive statistics for average age (years) of the parents at the birth of their offspring and generation intervals or generation length (years) for the four gametic routes in the Braford cattle breed. The average generation intervals or length for the historical population was 13.93 years, while the same parameter was 11.78 years for the current population (a stratified presentation of the results for the generation intervals for the four gametic routes in Braford cattle can be found in Appendix A). The average number of calves per mated bull decreased from 55.56 to 12.44 from the historical to the current population, while the average number of calves per mated cow increased from 1.75 to 6.39. The ratio of cows to bulls mated increased from 46.54/1 to 50.30/1. Bull selection intensity increased from 2.45% to 2.76% from the historical to the current population, while historical cow selection intensity almost doubled values in the current population (59.76% to 33.43%). The maximum progeny per bull (10,429) and cow (88) in the historical population moderately decreased in the current population to a number per mated bull of 7412 and for mated cows to 52 calves, respectively, as shown in Table 1. Figure 2 and Figure 3 describe the yearly evolution of censuses and diversity parameters in Braford cattle.

### 3.2. Inbreeding, Coancestry, and Degree of Nonrandom Mating

Table 3 presents the results for average *F*, ΔF, maximum *F*, inbred and highly inbred animals (%), *C*, ΔR, assortative or nonrandom mating rate (α), and GCI. The average *F* was low, and an increasing trend was reported (0.001% in the historical population and 0.002% in the current population) although highly inbred animals have historically been and currently are present in the population (maximum *F* of 33.26% and 26.66%, respectively). The percentage of inbred animals was 7.05% and 5.32%; the average *C* was 0.001% and 0.002%, and the degree of nonrandom mating (α) reached a progressively increasing value of −0.0001 to 0.0001 for the historical and current population sets, respectively (Table 3). The average *F* reached a 0.031% maximum in 1967, while the 0.003% maximum average coancestry was reached in 2017. The average degree of nonrandom mating reached a maximum of 0.03 in 1967, while its minimum was reached in 1951 (−0.002). In terms of matings between highly inbred animals, 3 (0.00%) matings between full siblings, 577 (0.16%) matings between half siblings, and 265 (0.07%) parent–offspring matings have occurred.

### 3.3. Probabilities of Gene Origin and Ancestral Contributions

The results for the analysis of gene origin probabilities, ancestral contributions, and genetic diversity loss are shown in Table 4 and Table 5. Considering the marginal genetic contribution, a single ancestor (identification number: 17,640) explained from 4.55% to 7.22% of the genetic pool of the animals with both parents known in the historical population (reference population) and also marginally contributed to 0.003% of total inbreeding and 0.008% of total coancestry. The contribution to the population genetic pool through nodal common ancestors forming inbreeding loops was 0.76%. The top 10 ancestors accounted for 0.024% of total inbreeding and 0.038% of total coancestry. The effective population size based on the individual inbreeding rate (*NeFi*) was 462.9630, while based on the individual coancestry rate (*NeCi*) was 420.1681 (through all the coancestries computed between animals of a different sex). The number of equivalent subpopulations was 0.9076.

A summary of measures of genetic diversity and analysis of gene origin, effective number of nonfounders (N_ef_), number of founder equivalents (f_e_), and effective number of ancestors (f_a_) are presented in Table 4 and Table 5.

### 3.4. Country, Province, and Owner Relationships

A summary of Wright’s F-statistics is shown in Table 6. The 224, 22, and 4 existing subpopulations were computed considering owners/farms, provinces, and countries as the subdivision criteria. The mean number of animals per owner/farm, province, and country was 1598.40, 16,274.59, and 89,510.25, respectively. A total of 24,976, 230, and 6 Nei’s genetic distances were considered, respectively, when owner/farm, province, and country were used as the differentiation criteria. Nei’s average genetic distance between owners/farms was 0.0036, 0.0006, and 0.0001, respectively, for owner/farm, province, and country. Mean coancestry within subpopulations was 0.0047, 0.0020, and 0.0015, respectively, when owner/farm, province, and country were considered as the subdivision criteria, while the mean coancestry values in the metapopulation were 0.5005, 0.5006, and 0.5006, and self-coancestry values were 0.0012, 0.0013, and 0.0013, respectively, when owner/farm, province, and country were used as the differentiation criteria. Studying Wright’s F parameters, the *F* relative to the total population (*F*_IT_) were −0.0001, −0.0002, and −0.0002, and the *F* relative to the subpopulation (*F*_IS_) were −0.0036, −0.0008, and −0.0003, when owner/farm, province, and country were used as the differentiation criteria, respectively. The correlation between random gametes drawn from the subpopulation relative to the total population (*F*_ST_) was 0.0036, 0.0006, and 0.0001, respectively (Table 6).

The assessment of the herd, provinces, and country structures revealed none of them could be considered the population nucleus, meaning that breeders not only use their own males, but also purchase and sell them, hence, none of the herds, provinces, and countries could be considered to be completely isolated [48]. The number of owners/farms, who/which did not use their own bred sires (commercial and multiplier herds) was more than twice as low as the number of those that did, and none of the herds was totally isolated.

One pair of herds had the greatest Nei’s genetic distance between them (0.191), while the closest pair of herds were 0.019 apart. The longest distance between provinces was between Sidney and Victoria, both in Australia, while the shortest one (0.002) was between Santa Fe and Corrientes in Argentina.

Uruguay and Australia were genetically the most distantly related countries, while Paraguay and Argentina were those countries between which the shortest distance was held. The mean Nei’s minimum distance/average homozygosity was 0.0036, 0.0006, and 0.0001 across owners/farms, provinces, and countries, respectively. Dendrograms displaying the relationship among owners/farms, provinces, and countries after the computation of Nei’s genetic distances are shown in Figure 4, Figure 5 and Figure 6.

## 4. Discussion

### 4.1. Census Evolution, Herd Number, Generation Intervals, and Bull to Cow Ratio

The number of herds of Braford cattle, as well as the number of provinces and animals per herd, has historically decreased up until the current period (Figure 2). This situation responds to the important changes that the livestock area suffered in some of the countries under study over the years, mainly in Argentina and Uruguay (Table 1) [49]. These structural and geographical changes can be mainly ascribed to the growing expansion of the agricultural sector. Contextually, the growth of soybean cultivation in Argentina during the last decade caused livestock to reduce its utilizable surface by more than 15 million hectares, which forced a territorial rearrangement. Similar events, although of lesser magnitude, occurred in Uruguay, where livestock lost more than 900,000 hectares per year in favor of the cultivation of soybeans in the last decade [50]. By contrast, in Paraguay, there was an increase in production based on a growth in the stock due to the possibility of the expansion of livestock frontiers by resorting to clearing forest regions [51].

Another possible cause for the decrease in the number of animals and herds could be attributed to the drought suffered by some of the countries studied, such as Argentina, which impacted on the country’s livestock systems between 2008 and 2009, and which translated into an overload of animals and a lack of food [52].

The impact of drought periods on the effective number of animals lost was felt most in those areas where a rather drastic hydric deficit had been reported. La Pampa, Chaco, north of Santa Fe, and Corrientes were the regions where higher losses were registered, even if these were the areas where a higher population increase was reported during the last 14 years with values of 40%, 52.9%, 29.2%, and 34.3%, respectively [49]. A similar event was described in Australia from 2000 to 2010, where droughts caused an effective loss of 27 million cattle [53].

The number of Braford births remained stable from 1949 to 1967 when the first population peak was detected. This peak not only coincided with the introduction of the first individuals in Latin America [3] but also with the worldwide expansion of the breed to countries such as South Africa around the 1970s (1975 was when the first attempt of a South African breeding program would take place). Most of the first introduction movements of the breed were independently-carried crosses by herdsmen rather than crossbred individual importations from the USA or Australia. Afterward, the number of births was constant until an increase was detected around the year 2004, which went hand in hand with the creation of the World Braford Confederation in Houston, Texas, USA in 2001 (integrating Australia, Brazil, Paraguay, the USA, Uruguay, and Argentina), which marked the most relevant moment of the global expansion of the breed [7].

The number of bulls and cows described opposite trends, as a historical increase in the number of bulls and a decrease in the number of cows were reported up until the present. This finding may be supported by the fact that the Braford breed is mainly derived for meat production; hence, the interest in bulls is much greater than the interest in cows. Such an interest was the causative agent of the increase in the proportion of males in the population throughout history as a consequence of the increase in the reposition of bulls and of the commercialization of cows, as reported by most of the properties.

Historical generation intervals were over the mean found in other breeds, such as Simmental [54], Romosinuano [55,56], Brahman [57], Gir, Guzerá, Indubrasil, Nelore, Sindi, and Tabapuã zebu [58,59]; and Angus, Devon, Hereford, and Shorthorn [60]. Even though the generation interval decreased in the current population with respect to the historical one, it remains long. This interval could have been used in an efficient manner to prevent an increase in breeding, as it may translate into a longer time needed to obtain reproductively active animals.

The mean generation intervals increased for the gametic routes of bull to son and cow to son, while they decreased in the routes of bull to daughter and cow to daughter. This was probably due to a matter of handling, which conferred a longer period for males to achieve a better reproductive performance, from the perspective of service ability and semen quality. However, intervals from either bull or cow to daughters were the longest ones that have also been found in breeds such as Angus, Devon, Hereford, and Shorthorn bred in Brazil [59].

The opposite was found in Angus, Black Brangus, Red Brangus, Hereford, Limousine, Salers, and Braunvieh, for which the mean generation intervals were longer in the route of bulls to sons than in any other of the other gametic routes, which suggests the use of the same bulls during extended periods of time together with a reduced number of males being used and the use of artificial insemination [61].

The results for each of the four gametic routes may provide a way to infer whether paternal and maternal lines within a breed are being handled in the same manner within herds and what their relative contributions to the population have been over time. Long generation intervals could mainly be attributed to a slow rotation rate, as the most favored and popular bulls and cows continued contributing to their offspring in posterior generations for years.

Contextually, longer generation intervals in maternal lines are often found in the bovine species, especially when breed handling determines a longer stay of cows within their herds. Elongating generation intervals can be useful to increase the number of bulls and cows selected for breeding, progressively increasing the population’s effective size, which is inversely proportional to the inbreeding coefficient.

The number of calves per mated bull was reduced from the historical population to the current population, while the opposite occurred in the case of females, in which the number of calves per cow increased. These results may derive from the policies, which indeed gave way to the Australian Braford ‘line’ and which may have continued over time and transferred to South American populations during the process of internationalization. The Brahman was regarded as the breed to crossbreed to the Hereford in order for the latter to achieve features, which were poorly developed. In the Australian ‘line’, the Hereford cows participated in the cross contrary to what happened in the (North) American ‘line’. These results denote the appreciation of females, which is also reflected in the historical number of females doubling the number of males. Afterward, the increasing implementation of reproductive biotechnologies used by producers, such as the use of embryo transfer, may have translated into an increase in the number of offspring per cow [62]. Such findings compare to those by Ramírez-Valverde et al. [61] in Angus, Black Brangus, Red Brangus, Hereford, Salers, and European Swiss for the number of calves per bull, while the number of calves per cow observed in the current population were lower than those of the present study.

The relationship between mated bulls to cows increased from the historical period to the present; this may be due to a better selection of bulls by the owners as regards seminal characteristics, testicular biometry, sexual behavior, and phenotypic or morphological characteristics. When these criteria are taken into account, it is possible to select the best bulls from the herd and thus decrease the number of bulls per cow which can translate into functional and economic benefits [63].

On the other hand, the intensity of the selection of bulls slightly increased in the current period, while that of cows decreased almost by half. This fact coincides with the increase in the number of bulls from the historical period to the present, since as the availability of bulls increases, the possibilities to increase the intensity of selection and choose the best males depending on the personal interests of the producer increases as well. At the same time, the number of cows decreased over the years, which led to a reduction in the intensity of their selection, as the possibility to perform a rather strict selection declines with the number of animals.

The maximum progeny per bull was always high, both in the historical period and in the present; however, it decreased over the years, while the maximum progeny of cows decreased in the current period, perhaps due to the same reasons for which the general population decreased, as aforementioned.

According to literature [3,4,7], a strong point of the Braford is the breeding female with her excellent reputation for fertility, and her ability to rear a top vealer. Combine this with her reputation for ease of calving, and you have one very productive female. The mating age for females is generally from 15 months old, depending on the conditions in which they are kept. Young bulls are on average capable of working in a commercial situation from 18 months to 2 years old onwards [64].

### 4.2. Inbreeding, Coancestry, and Degree of Nonrandom Mating

Inbreeding was always low, both in the historical period and in the current one, with results that are similar to those observed by Piccoli et al. [60] in Brazil for Angus, Devon, Hereford, and Shorthorn breeds. However, in the present study, there was a peak in inbreeding of 0.031% around 1967, coinciding with the time of the introduction of the breed in Latin American countries such as Brazil, Argentina, Paraguay, and Uruguay, which may probably be ascribed to the fact the first crosses of the specimens were performed between animals who were closely related to each other. After that peak, the values decreased to almost 0.00% until the current period. Higher values of inbreeding were observed by Ramírez-Valverde et al. [61] in Mexico for bovine meat breeds, reporting low values between 0.9% and 3.5%.

Higher levels (F > 4%) of inbreeding have been reported for certain populations, which could potentially be attributed to the overuse of the few mating bulls. However, inbred animals were always present in the population, given the maximum inbreeding coefficient found was 33.26%, similar to that observed by Cavani et al. [57], who reported a maximum inbreeding coefficient of 40.62% in males of the Brazilian Brahman.

The percentage of inbred animals was 7.50% to 5.32%, a reduction from the historical period to the present, respectively. This is favorable, and probably due to better planning of matings between the animals, using the least related ones as parents. This is also reflected in the value of the degree of nonrandom mating (α), which increased from the historical period to the current one. This should be regarded as a positive sign, as it shows that selection practices used by herd owners are focusing on mating less related animals in the crosses. This value of α remained constant throughout the years, having a peak around 1967, coinciding as previously mentioned with the introduction of the animals to Latin America, as initially there were fewer specimens, which promoted the fact that the animals interbred randomly, but among presumably related individuals, thus increasing the inbreeding levels in the population.

The values of coancestry observed in the animals of the Braford breed were low both in the historical period as well as in the current one, and they were lower than those observed by de Faria et al. [58] in the Brazilian Brahman (3.6% and 4.8%). The values were kept constant, with some observed peaks during the years 1992 and 2001, and a maximum that was reached in 2017. This finding suggests breeding controls were less strict, thus mating between related animals appeared more frequently.

The average individual increase in inbreeding (ΔF) was 0.0004% in the historical period but doubled in the current period to 0.0008%, while in the Angus, Black Brangus, Red Brangus, Hereford, Limousine, Salers, and European Swiss breeds in Mexico the observed values were 0.32%, 0.74%, 1.04%, 0.65%, 0.26%, 2.08%, and 0.45%, respectively [61]. Lower values may derive from the fact that the Braford, given its inner condition of being a crossbred population, is naturally prone to use animals that as a basis are not reciprocally related.

The average relatedness coefficient (ΔR) in this study was 0.002% in the historical period, and it increased in the current period to 0.004%, which was still much lower than that observed in the Romosinuano breed in Mexico, which was 3.23% by Hidalgo et al. [56] and 0.99% by de Araujo Neto et al. [65] and lower than that observed by Piccoli et al. [59] in the Brazilian Angus, Devon, Hereford, and Shorthorn with values ranging from 0.25% to 2.42%. For the Braford population under study, the ΔR was kept practically constant through the years with a peak occurring in 2017 of around 0.006%. Although the ΔR decreased between 1979 and 1989, and an increase in 1992 was identified, these were not substantial in the context of the ΔR average of around 0.002% ± 0.002%.

The relatedness coefficient is inversely related to genetic diversity and can be used as a long-term indicator of inbreeding (F) evolution. When the ΔR is greater than F in the population, mating between relatives is more frequent and, in general, when the ΔR tends to approach zero, genetic diversity increases. Therefore, when selecting the best animals, it is important to consider the animals with the lowest ΔR values.

The Genetic Conservation Index (GCI) increased from the historical period to the present, from 2.32 to 3.13. These values are lower than those observed by Hazuchová et al. [66] in Slovak Spotted bulls, for whom the GCI was 4.18. The ideal individual would receive equal contributions from all the founder ancestors in the population and, consequently, would wholly represent the gene pool of the founder population. In this context, the higher the values of the GCI, the more diverse the animal in question is and the higher its value for conservation.

### 4.3. Probabilities of Gene Origin and Ancestral Contributions

The number of founders that contributed to the reference population in the present study was 89,743, while the number of ancestors was 86,329. These values are highly variable depending on the bovine population under study. For instance, the study by Ramírez-Valverde [60] reported that the number of founders in the populations of Angus, Black Brangus, Red Brangus, Hereford, Limousine, Salers, and European Swiss was 10,168, 4827, 2131, 2439, 5862, 1728, and 11,886 respectively, while the number of ancestors was 224, 33, 55, 254, 199, 105, and 166, respectively.

Núñez-Domínguez et al. [55] reported a value of 183 to 827 founders in the Romosinuano breed. The number of founders in the present study exceeds even those reported in other highly selected breeds, such as the Mexican Charolais with a value ranging between 5000 and 13,000 founders [67]. The effective number of nonfounders (N_ef_) in this work was 490,067, about seven times higher than the value reported for the Simmental breed (68,400) [64]. The equivalent number of founders (*fe*) was 2944.77. These values are higher than those observed by Núñez-Domínguez et al. [55] in six bovine breed populations for which it ranged between 113 and 541 and was markedly higher than the Romosinuano breed, for which the values ranged between 50 and 60 [55]. The number of founders was higher than *fe*, which was indicative of a reduction in genetic diversity, which may be ascribed to an imbalanced use in the number of founders. The effective number of ancestors (*fa*) was 162 and higher than the values reported for the Romosinuano breed [55], which ranged between 24 and 31, while in the aforementioned six breeds in Mexico, these values ranged between 33 and 254 [61], which were similar to those in the Charolais, which ranged between 207 and 247 [67].

In the present study, the estimated *fa* in the seven bovine populations analyzed was lower than their respective *fe*, which indicates a decrease in genetic diversity due to the presence of genetic bottlenecks in all population sets from their foundation. This imbalance in the values of *fa* and *fe* has been reported for breeds suffering a process of internationalization from a specific point of origin. From a genetic diversity perspective, the relocation or reintroduction of a section of a breed to a new place to serve as the founder basis of a new population may produce a similar population fragmentation effect as a sharp reduction in the original population (if the breed suddenly became endangered), given a relatively limited number of animals may serve as the population basis on which the international expansion of such breed will be built [27,68]. Nonetheless, this effect is somehow buffered when the breed in particular is made out of a crossbreeding process between two or more breeds, as it occurs in the Brafords. Furthermore, in the case of the Brafords, this buffering effect is specially reinforced because in South America different percentages of blood of each of the participating breeds were tested and permitted in the standard seeking the particular economic interest of each country.

Such a buffering effect is denoted by the fact that, in the current study, the number of founder genome equivalents (*fg*) or the number of equally-contributing founders that would be expected to produce the same genetic diversity as observed in the current population if there is no random loss of founder alleles in descendants (e.g., through genetic drift) was as high as 420.154.

These values were much higher than that observed by Ríos-Utrera et al. [68] in the Charolais in Mexico, for which values ranged between 127 and 143 from 1984 to 2018 and were also higher than those observed for six bovine breeds (Angus, Black Brangus, Red Brangus Rojo, Hereford, Limousine, Salers, and European Swiss) in Mexico, which ranged between 19 and 361 [61].

The ratio between *fa/fe* was 0.06, which was much lower than that observed in Angus, Black Brangus, Red Brangus, Hereford, Limousine, Salers, and European Swiss, for which values were 41.4, 29.2, 28.6, 61.5, 38.3, 31.7, and 54.1, respectively.

Cavani et al. [57] reported a value of 1 for *fa/fe* across all the years that their study lasted in a population of the Brahman in Brazil, which may be indicative of a lack of bottlenecks and relatively low losses due to genetic drift. The *fg/fe* ratio in the aforementioned study was also close to 1, while this value decreased to 0.14.

The ratio of *fg/fe* measures the magnitude of the genetic drift such that the lower *fg/fe*, the greater the effect of genetic drift. In the present case, *fe* is larger than *fg*, which denotes the fact that some mating bulls are underused when compared to others; hence, only those with a larger number of descendants are able to preserve their genome along generations. The number of ancestors that contributed to explaining the 50% of founding genes in the population in the present study was 317, which surpassed the six breeds studied in Mexico in the study by Ramírez-Valverde et al. [61], which ranged between 16 and 177. These contributions of ancestors to the genes of the respective populations should be considered by breeders, because the higher presence of genes of a few mating animals in the populations could lead to an increase in matings between related animals and, consequently, to a generation of inbred individuals and a loss of genetic variability.

### 4.4. Country, Province, and Herd Relationships

The values of the coefficient of inbreeding relative to the total population (F_IT_) was −0.0001, −0.0002, and −0.0002, and the F relative to the subpopulation (F_IS_) was −0.0036, −0.0008, and −0.0003 when the owner/farm, province, and country were used as the differentiation criteria, respectively. The correlation between the random gametes drawn from the subpopulation relative to the total population (F_ST_) was 0.0036, 0.0006, and 0.0001. Similar values were observed in Afrikaner cattle from South Africa, where the unbiased estimates of Wright’s F-statistics were 0.027 for F_IT_ and −0.022 for F_IS_. F_IS_ was slightly negative, indicating a small surplus of heterozygotes within herds, while F_IT_ was slightly positive, indicating a small overall surplus of homozygotes throughout the population [69]. Studies by Rovelli et al. [70,71] reported similar values to those found in the present study of 0.005, −0.011, −0.023, −0.023, and −0.030 for F_IS_ and 0.085, 0.076, 0.079, 0.074, and 0.073 for F_ST_ in cattle from the Marchigiana, Chianina, Romagnola, Maremmana, and Podolica breeds, respectively.

According to literature, the average F_IS_ value was relatively low for the Chianina, Romagnola, Maremmana, and Podolica breeds, indicating a positive effect of controlled genetic inbreeding in the breeds [70], whereas it was slightly positive for Marchigiana (0.005). The low values of heterozygosity and the high inbreeding of Marchigiana compared with other breeds are attributable to the extensive use of a small number of improved bulls [72]. The extensive use of artificial insemination in the Marchigiana, Chianina and Romagnola breeds could be responsible for lower values of heterozygosity compared to the two heritage breeds. Thus, a responsible use of the mating plans in these breeds is recommended to avoid the loss of variability and the increase of inbreeding [73]. The effective population size based on the individual inbreeding rate (*NeFi*) was 462.96 in the present work, while based on the individual coancestry rate (*NeCi*) it was 420.17. Santana et al. [59] observed *NeFi* values of 158.5, 104.1, 118.1, 40.3, 100.23, 37.4, and 101.5 and *NeCi* of 151.5, 138.6, 192, 199.6 113.33, 84.2, and 169.2 for the Brahman Gir, Guzerá, Indubrasil, Nelore, Sindi, and Tabapuã zebu breeds from Brazil, respectively. Table 5 shows the contribution of the ancestors in the different percentages of the gene pool: 25% of the pool is explained by 17 animals, 50% by 317, 75% by 15,436, and 100% by 86,297 animals. These results are higher than those observed by Santana et al. [58] in animals of the Brahman, Gir, Guzerá, Indubrasil, Nelore, Sindi, and Tabapuã zebu breeds from Brazil, for which between 3 and 6 animals explained 25% of the gene pool, between 11 and 32 explained 50%, and between 31 and 227 explained 75% of the pool.

In the European context, 30% of the gene pool was explained by 4 animals, 50% was explained by 13 animals, 70% by 36, and 90% by 751 animals, in a Braunvieh population between 1990 and 2014 [73]. Table 4 and Figure 7 and Figure 8 show the percentage of genetic diversity (GD), which is 99.88%; the loss of genetic diversity is 0.12%; the loss of genetic diversity due to bottlenecks is 0.12%, and the loss of genetic diversity due to unequal contributions from the founders is 0.02%. These results are similar to those observed by Wirth [74] in the aforementioned Braunvieh animals, in which the loss of genetic diversity due to bottlenecks was 0.025%, due to genetic drift was 0.017%, and due to the unequal contributions of the founders was 0.008%.

The evaluation of the structure of the herds, provinces, and countries revealed that none of them could be considered the nucleus of the population. The number of owners/farms that did not use their own bulls was more than twice the number that did, and none of the herds was totally isolated. One pair of herds had the greatest Nei genetic distance between them (0.191), while the closest pair of herds was separated by 0.019. Sydney and Victoria were the most genetically distant provinces in the study, even if they are both in Australia and relatively close (950 km). This was indirectly indicative of the efficiency of the breeding programs implemented, since the least related animals were carefully studied and chosen to perform matings. The shorter Nei genetic distance, observed between Santa Fe and Corrientes, could be ascribed to the fact that both neighboring Argentinean provinces exchange genetic material from their best bulls, and the commercial activities involving the Braford breed animals is important. Uruguay and Australia were genetically the most distantly related countries, even though both have continuous import and export flows of Braford animals, and the semen, embryo and live animal commercialization between both countries is constant. The shortest genetic distance between Paraguay and Argentina could be attributed to the fact that neighboring countries work in conjunction with breeding programs for Braford breed, continually exchanging genetic material.

## 5. Conclusions

The growing expansion of the agricultural sector that the livestock area suffered in some of the countries under study over the years, mainly in Argentina and Uruguay, and the drastic drought periods are responsible for the decreasing trend of the breed’s censuses. The creation of the World Braford Confederation marked the moment of the global expansion of the breed. The interest for bulls is much greater than the interest for cows due to the meat production profile of the Braford breed. Long generation intervals could have been used as an efficient manner to prevent the increase in breeding, as they may translate into a longer time needed to obtain reproductively active animals. Elongating generation intervals can be useful to increase the number of bulls and cows selected for breeding, progressively increasing the population’s effective size, which is inversely proportional to the inbreeding coefficient. The number of calves per mated bull decreased due to the increasing implementation of reproductive biotechnologies used by producers, such as the use of embryo transfer. Less strict breeding controls may make matings between related animals appear more frequently, although matings are naturally prone to occur between animals that are not reciprocally related given that the Bradford is a crossbred population. The presence of genes of a few mating animals in the populations could lead to an increase in matings between related animals and, consequently, to a generation of inbred individuals and a loss of genetic variability. However, a certain gene flux is still maintained from Australia to South American countries as supported by the analysis of genetic distances. Our results may provide the basis to tailor specific strategies to be implemented internationally to ensure the preservation of genetic diversity in Braford cattle. In this regard, measures, such as the use of artificial insemination or embryo vitrification, need to be reinforced to prevent inbreeding rate increases and to increase the effective population size through the connection of herds. Furthermore, the evaluation of the genetic relationships shared between the cow and bull comprising each breeding pair may permit selecting individuals for mating when acceptable offspring coancestry levels are ensured. This may be additionally enhanced by working on the male side and focusing on reducing the overuse of specific males and using bull rotation policies to limit their genetic repercussions in herds to ensure the Braford herds enjoy proper genetic health.

## Figures and Tables

**Figure 1 animals-12-00275-f001:**
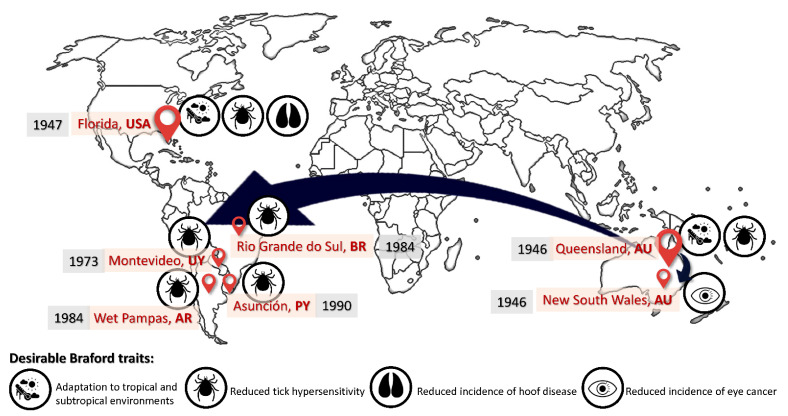
Time map of the causes for the historical distribution of the Braford breed from their original focuses in Australia and the USA into South America.

**Figure 2 animals-12-00275-f002:**
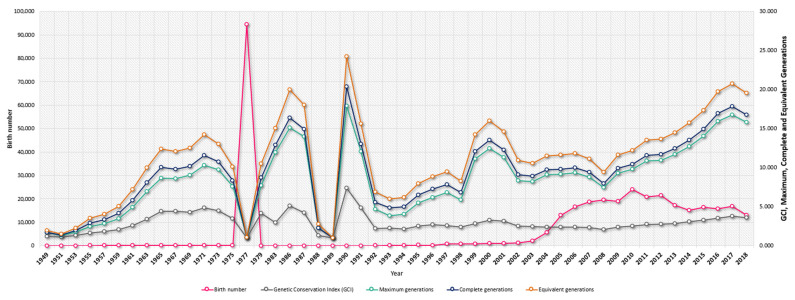
Yearly evolution of Braford cattle birth numbers, Genetic Conservation Index (GCI) and Maximum, Complete, and Equivalent Generation numbers from 1949 to 2019.

**Figure 3 animals-12-00275-f003:**
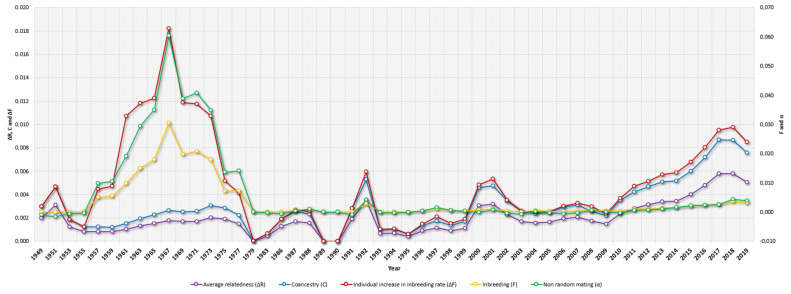
Yearly evolution of Braford cattle average relatedness (ΔR), Coancestry (C), Individual Increase in Inbreeding (ΔF), Inbreeding (F), and Nonrandom mating (α) from 1949 to 2019.

**Figure 4 animals-12-00275-f004:**
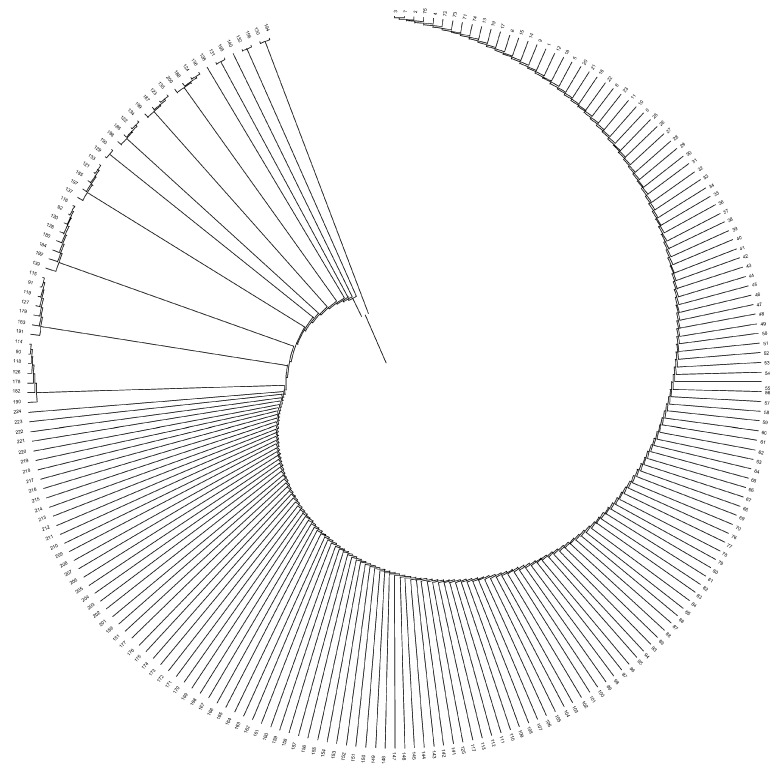
Dendrogram displaying owners/farms after computing Nei’s genetic relationships.

**Figure 5 animals-12-00275-f005:**
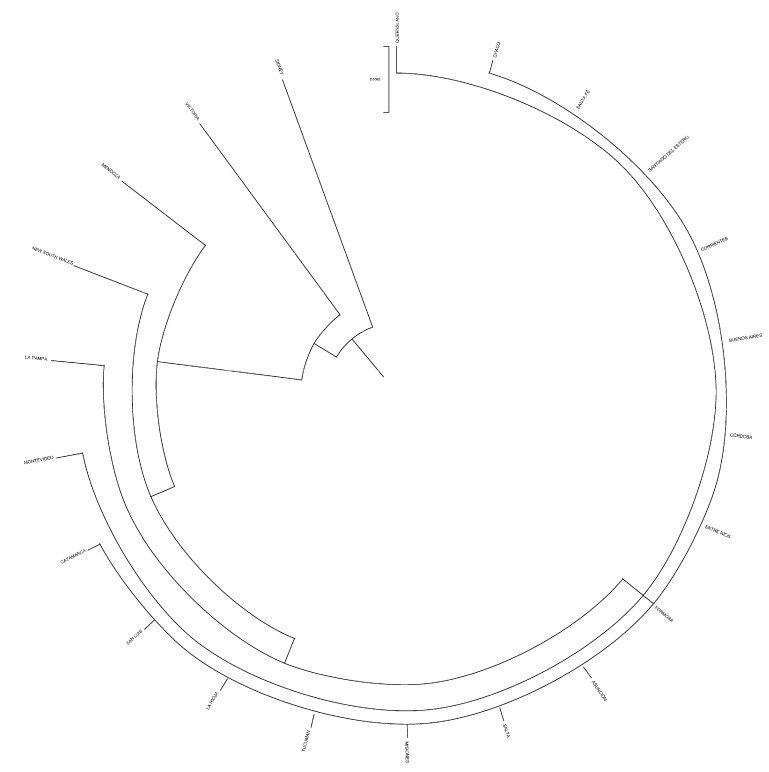
Dendrogram displaying provinces after computing Nei’s genetic relationships.

**Figure 6 animals-12-00275-f006:**
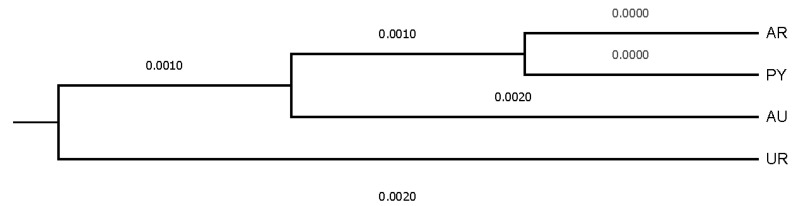
Dendrogram displaying countries after computing Nei’s genetic relationships.

**Figure 7 animals-12-00275-f007:**
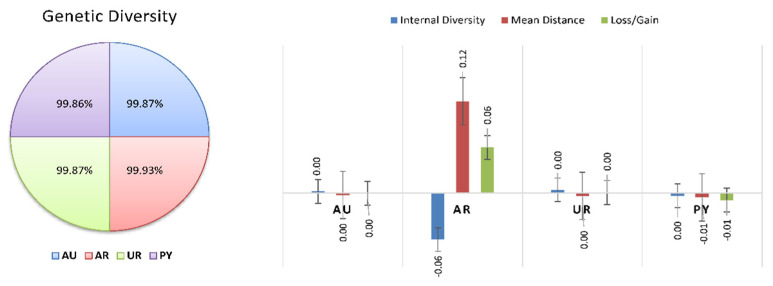
Genetic diversity and diversity loss in Braford cattle breed across countries.

**Figure 8 animals-12-00275-f008:**
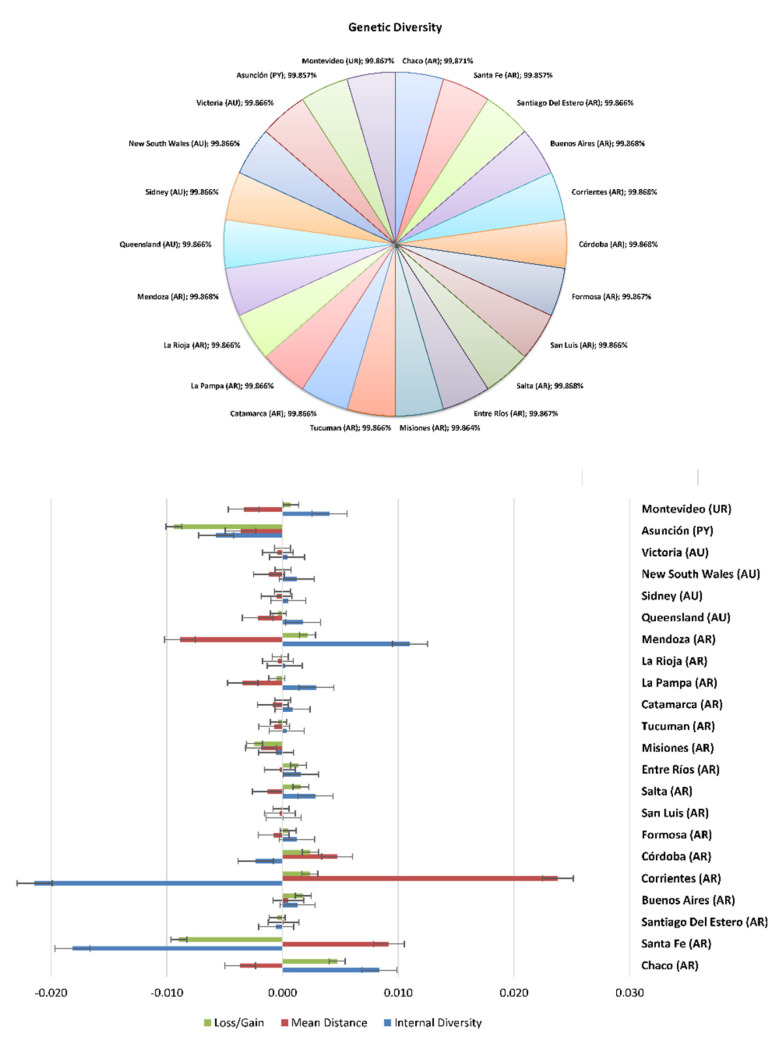
Genetic diversity and diversity loss in Braford cattle breed across provinces.

**Table 1 animals-12-00275-t001:** Summary of demographic and breeding-related statistics.

Parameter/Population Set	Historical	Current
Total number of herds	223	180
Total number of provinces	22	16
Total number of countries	4	4
Average number of animals per herd/average herd size	1605.57	643.10
Total bull percentage %	34.83	46.45
Mean number of calves per bull, n (artificial insemination included)	1.36	0.34
Maximum number of calves per mated bull, n (animals with unknown sire excluded/artificial insemination included)	10,429	7412
Mean number of calves per mated bull, n (animals with unknown sire excluded/artificial insemination included)	55.56	12.44
Average age of bull in reproduction, years	7.38	7.43
Total cow percentage, %	65.17	53.55
Mean number of calves per cow, n (embryo transfer included)	1.04	2.07
Maximum number of calves per mated cow, n (animals with unknown dam excluded/embryo transfer included)	88	52
Mean number of calves per mated cow, n (animals with unknown dam excluded/embryo transfer included)	1.75	6.39
Average age of cows in reproduction, years	8.58	9.16
Total Cow/Bull Ratio	1.87/1	1.15/1
Mated Cow/Bull Ratio	45.64/1	50.30/1
Progeny from bulls selected for breeding, %	15.96	3.31
Progeny from cows selected for breeding, %	41.43	40.96
Male selection intensity or portion of male calves born retained for breeding, %	2.45	2.76
Female selection intensity or portion of female calves born retained for breeding, %	59.76	33.43

**Table 2 animals-12-00275-t002:** Summary of statistics of population completeness level.

	Population Set	Historical	Current
Parameter			
Population size		358,041	115,757
Maximum number of traced generations, n		19	19
Pedigree completeness level at 1st generation, (Known parents)		57.72	76.20
Pedigree completeness level at 2nd generation, (Known grandparents)		27.10	46.48
Pedigree completeness level at 3rd generation, (Known great grandparents)		16.91	29.50
Pedigree completeness level at 4th generation, (Known great great grandparents)		11.12	17.66
Pedigree completeness level at 5th generation, (Known great great great grandparents)		9.02	13.37
Number of maximum generations (mean ± SD)		6.22 ± 7.04	6.22 ± 7.04
Number of complete generations (mean ± SD)		0.55 ± 0.68	0.55 ± 0.68
Number of equivalent generations (mean ± SD)		1.50 ± 1.57	1.50 ± 1.57

**Table 3 animals-12-00275-t003:** Summary of pedigree analysis statistics.

	Populational Sets	Historical *n* = 358,041	Current *n* = 115,757
Parameter			
Inbreeding coefficient (F, %) (mean ± SD)		0.001 ± 0.010	0.002 ± 0.014
Average individual increase in inbreeding (ΔF, %) (mean ± SD)		0.0004 ± 0.0067	0.0008 ± 0.0067
Maximum coefficient of inbreeding (%)		33.26	26.66
Inbred animals (%)		7.50	5.32
Highly inbred animals (%)		0.26	0.49
Average coancestry coefficient (C, %) (mean ± SD)		0.001 ± 0.002	0.002 ± 0.002
Average relatedness coefficient (ΔR, %) (mean ± SD)		0.002 ± 0.004	0.004 ± 0.004
Nonrandom mating rate (α) (mean ± SD)		−0.0001 ± 0.010	0.0001 ± 0.0136
Genetic Conservation Index (GCI) (mean ± SD)		2.321 ± 1.894	3.135 ± 2.318

**Table 4 animals-12-00275-t004:** Measures of genetic diversity and genetic diversity loss.

Parameter	Reference Population (Both Parents Known) (*n* = 14,538)
Genetic diversity, GD (%)	99.88
Genetic diversity loss, GDL (%)	0.12
Genetic diversity in the reference population considered to compute the genetic Diversity loss due to the unequal contribution of founders, GDL (%)	99.98
GDL due to bottlenecks and genetic drift since founders (GL) (%)	0.12
GDL due to unequal founder contributions (%)	0.02

**Table 5 animals-12-00275-t005:** Probabilities of gene origin and founder analysis.

Parameter	Reference Population (Both Parents Known) (*n* = 14,538)
Historical population	358,041
Current population	115,757
Base population (one or more unknown parents)	194,109
Actual base population (one unknown parent = half founder)	151,378
Number of founders contributing to the reference population, *n*	89,743
Number of ancestors contributing to the reference population, *n*	86,329
Effective number of nonfounders (*Nef*)	490.08
Number of founder equivalents (*f_e_*)	2944.77
Effective number of ancestors (*f_a_*)	162
Founder genome equivalents (*f_g_*)	420.15
*f_a_/f_e_ ratio*	0.06
*f_g/_f_e_ ratio*	0.14
Ancestors explaining 25% of the gene pool (*n*)	17
Ancestors explaining 50% of the gene pool (*n*)	317
Ancestors explaining 75% of the gene pool (*n*)	15,436
Ancestors explaining 100% of the gene pool (*n*)	86,297
Average individual increase in inbreeding (ΔF) (%)	0.02
Average relatedness (ΔR) (%)	0.33

**Table 6 animals-12-00275-t006:** Wright’s Fixation statistics and heterozygosity parameters when subdivision criterion is the breeder, province, and country of origin.

Parameters	Breeder	Province	Country
Number of predefined subpopulations	224	22	4
*F*_IS_ (Inbreeding coefficient relative to the subpopulation)	−0.0036	−0.0008	−0.0003
*F*_ST_ (Correlation between random gametes drawn from the subpopulation relative to the total population)	0.0036	0.0006	0.0001
*F*_IT_ (Inbreeding coefficient relative to the total population)	−0.0001	−0.0002	−0.0002
Mean inbreeding within subpopulations	0.0011	0.0012	0.0012
Mean number of animals per subpopulation	1598.40	16,274.59	89,510.25
Number of Nei genetic distances	24,976	230	6
Average Nei genetic distance	0.0036	0.0006	0.0001
Mean coancestry within subpopulations	0.0047	0.0020	0.0015
Self-coancestry	0.5005	0.5006	0.5006
Mean coancestry in the metapopulation	0.0012	0.0013	0.0013

## Data Availability

Data will be made available from the corresponding author upon reasonable request.

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
