# Peer review of "Process of Introduction of Australian Braford Cattle to South America: Configuration of Population Structure and Genetic Diversity Evolution"

_animals, 2022, doi:10.3390/ani12030275_

Round 1

Reviewer 1 Report

Word choice, grammar, sentence structure and general use of English. Overall, the quality of writing and use of English in this manuscript is excellent and only minimal editing is needed. Here are a few suggestions to improve the clarity and flow of your manuscript:

Word choice or omission. Occasionally, you select English words that are confusing within the context, or omit a key word, thereby changing the meaning of your sentence.

Examples from your simple summary: "...genetic diversity was conducted and shaped over the years." I think what you mean is: selective breeding was conducted, thereby shaping genetic diversity within South American Braford cattle. Selective breeding can influence genetic diversity, but it can't conduct it. Similarly, it would be less confusing in the last sentence of the abstract to say, "...the parameters of genetic diversity to suggest effective strategies for Braford breeders."

Example from your Discussion (line 524-525). "Planification" is not a word. Try a simpler approach, "This is favorable, and probably due to better planning of matings...". Shorter, clearer, tighter writing.

Sentence structure/order of clauses. Sometimes your sentence structure, particular word order, obscures your meaning or makes the sentence ambiguous. The first sentence of your abstract is a good example. I think you mean, "This study analyzes...of Braford cattle in South America from 1949-2019 to suggest effective strategies for breeding in the future." Clearly you're not suggesting breeding strategies to use in the past, but in the future, yet your current sentence is structured as if you are making recommendations for the past.

Omitted words. Example from your abstract, "Even if adequate levels of diversity can be found, practices that consider the overuse of bulls..." Obviously a bull (or at least his semen) must be involved in any breeding effort; what you really mean is overuse of individual bulls, which clearly can lead to increases in F.

I realize these comments a very picky, but they are truly meant to help. Precise communication is critical in population genetics- even more so than in many other science disciplines, especially in an important applied study like yours!

Introduction: the first paragraph is confusing. See above comments and apply to this paragraph. For example, you mean historical records, not theory. Partin & Hudgins breeding vs. Alto Adams Jr.???

Evolution, adaptability, environment, and artificial selection: Use these terms with care! In the context of evolution, adaptation is a change in genetic structure or allele diversity in a population. The ability of a living individual to tolerate or adjust to conditions its environment is not adaptation in this context. When you mean the living animals fail to thrive (i.e., slow growth rate, low productivity, high parasite load), you can say the breed is not adapted to its current environmental conditions OR that it is not genetically suitable to its current environmental conditions. An individual organism cannot adapt evolutionarily.

Evolution: do not use evolution to mean the history of something (e.g., history of cattle associations, breeding programs). Given your focus on genetic diversity and structure, use evolution ONLY to mean the change in allele frequencies within a population, and/or the genetic structure of the Braford metapopulation in South America.

Artificial selection: Unless herds are feral, with multiple cows and bulls roaming and mating freely without human influence, ALL domestic breeding events are artificial selection- particularly if straws of semen are used via AI. Use the phrase selective breeding instead.

Do not say "natural environmental conditions" unless you're contrasting with unnatural conditions (e.g., irrigation, air conditioning). Just say environment (the place) or environmental conditions (temperature, humidity, solar radiation, etc.).

Last sentence of first paragraph: I think you mean to say, "Brahman and Hereford crossbreeding occurred in an attempt to improve biological performance of cattle." The last clause is confusing. Do you mean the hybrid Braford cattle were not economically viable, or that the Hereford cattle were not??

Fig. 1: Major issues with your labels. (i) the desirable Braford trait is NOT climate adaptability, it's adaptation to tropical and subtropical environments., i.e., breeders sought cattle crossbreedings whose progeny would immediately perform well in Australia, SE U.S., etc. (ii) tick hypersensitivity- not sensibility. (iii) hooves always have some condition (excellent, poor, satisfactory). Be specific: which hoof ailments need to be reduced? (iv) very close! Use reduced incidence of eye cancer.

Change the wording of these key phrases throughout the manuscript.

INTRODUCTION

While it's okay to present a concise history of the breed in Australia and its introduction to South America, the rest of the history of associations, specific breeders, etc. is superfluous and confusing. Remove the detailing of the history of Braford cattle in the United States. Given the distinct development of these two populations of Brafords (e.g., use of Brahman cows vs. bulls to develop the breed) and lack of U.S.-origin Brafords in South America, the U.S. breed is irrelevant to your study.

Restructure the introduction so that the advantages of introducing Brahman alleles to Hereford cattle to help combat eye cancer, biological performance in tropical/subtropical climates, etc., are addressed in more detail right away. These distinctions of Brahmans (and therefore, Brafords) are critical. Reduce the details about the history of study of Braford resistance to heat stress, eye cancer, ticks. Just outline the advantages of the breed, concisely, to show why they're important in South America.

THEN detail the history of the development of Australian Brafords and their introduction to South America. Delete the detailed history of the cattle associations (irrelevant), though include mention of the various studbooks and how they were structured in each association. Include background information about the genetic diversity of Braford cattle, i.e., what was known prior to your study.

What do you mean by population effectives? If you mean that the breeding programs resulted in several distinct effective genetic populations, then clarify.

THEN state your study objectives. This structure will reduce redundancy in your manuscript and improve organization and clarity.

The paragraph composed of lines 158-170 does not make sense. Convoluted & confusing. Rewrite the entire paragraph, and reorganize the ideas/sentences for better clarity and flow.

The paragraph composed of lines 172-178 also does not make sense. Was your only objective to develop a model? Didn't you also use it to analyze F (inbreeding), C (coancestry), relatedness coefficent, etc., for Brafords in South America? What do you mean by "analysis of pedigree completeness downwards?" Define precisely "the population" you are studying (line 174 and elsewhere). The last sentence is a long, confusing run-on sentence. Divide into 2-3 simpler, clearer sentences. That sentence brings up "market demands in geographically diverse frameworks." Your study is not investigating market demands and how they vary across nations or other geographic entities- just delete that. If you want to specify Braford TRAITS that you hope to identify a genetic basis for through your study, fine. Focus on the population genetics, the genetic structure and allelic diversity of S. American Braford cattle.

Given that you analyzed data only from the Asociasiόn Braford Argentina, clarify in your title and Introduction which population you investigated. Were all these cattle bred in Argentina? Table 1 indicates cattle from four countries were included in your study. Which nations does the ABA studbook include? Does it include ALL Brafords in all four nations, only cattle bred in Argentina, or ?? The Brafords across South America may form a distinct metapopulation, or the ASA cattle may form their own distinct metapopulation, restricted geographically to Argentina and/or the areas adjacent. These details are CRITICAL to include. [well... I finally found the information... in lines 313-314, in the Results. This information is fundamental for the METHODS, so include it there!]

METHODS

Who is your intended audience? If you are only targeting population geneticists, then your shorthand use of pop gen jargon and statistical metrics is acceptable. If you are also targeting a more general audience composed of cattle breeders, animal scientists, agricultural educators and researchers, then you must briefly define what an F-statistic is (etc.). Pop gen is a jargon-filled discipline, and many scientists- never mind ranchers/breeders, won't know a lot of those details.

If you analyzed studbook data to create Table 1, it should be placed in Results, and the techniques used to generate the summary metrics addressed in Methods- even though they are routine demographic analyses. It's unclear whether the ABA provided the metrics in Table 1, or whether you calculated or determined these details from information in the ABA studbook.

Your control procedures, using only fully known bull-cow breedings, are good, but are presented in a confusing way. Clarify and explain why your control calculations are a good comparison to better reveal the implications of your analyses using the whole data set (entire ABA studbook).

Your analysis of ancestral contributions and gene origins is excellent... but poorly explained. How did you determine which bulls (and cows? Or no cows included?) were classified as founders? Again, if your intended audience includes cattle breeders (or agricultural scientists who are not grounded in evolutionary biology and population genetics), you need to define and explain terminology like effective population size, unequal (genetic) contributions, heterozygosity, genetic drift, and genetic distance. If you expect only evolutionary biologists and population geneticists to read your article, then you need not define the many terms you include. Almost all of my undergraduate biology majors, and most of our masters-level graduate students, would be lost and confused- and they all study ecology, evolution and genetics. But population genetics and all the statistical procedures used to wring out information from large genetic data sets is rather specialized information; just a little bit of explanatory information/definitions would allow a huge potential audience to benefit from your work!

RESULTS

Tables 3 and 4 are very cool! I found your results fascinating... and useful for breeders who are knowledgeable in population genetics. These tables ought to be in your Results section, however. This is a general weakness of your manuscript: placing information in the correct section. Some of your Introduction is better placed in your Methods; some of your Methods belong in the Introduction or Results; some of your Results belongs in the Methods.

Please reorganize your article to put information where readers expect (and need) it to be.

The actual results are eye-opening and useful. Fewer, smaller herds? Not what I would have expected. Relatively more bulls contributing genetically, but fewer cows? Also unexpected- especially when the mean number of calves/cow increased so strongly. You put these outcomes clearly in context by investigating selectivity of bulls and cows, both of which increased. Figures 2 & 3 are very nice, but would benefit from a bit more information in the caption (or body of the Methods or Results) explaining what the Genetic Conservation Index, Coancestry (etc.) actually tell us about this population. Again, if you're targeting only pop gen experts, never mind.

The paragraph beginning line 366 is much too long and dense; break it up into 2-3 shorter, more focused paragraphs. These results are complex and need to be digested in smaller chunks.

The fact that NONE of the individual farms, provinces or nations can be considered a genetic nucleus to these Braford cattle seems a critically important point, but it's deeply buried in the above, long paragraph (lines 382-384).

Wait... I'm confused. Your genetic distances analyses include populations from Australia? That wasn't clear before. Be sure each of the inclusions and exclusions from each of your analyses is clearly stated in the Methods, and not deeply buried. Your genetic distances outcomes are fascinating, however! I very much like Figures 6, 7 and 8. It was fun to peruse them in detail, and they support your narrative very well. It would be nice to have the individual farm, its location, and and province locations provided an Appendix listing the ID used in the genetic-distances figures. for each farm and province so readers can figure out which comparison each branch of the dendrogram represents. It's obvious for the nations, but not for the individual farms, their geographic locations, or those of the provinces.

DISCUSSION

Overall, I like your discussion. The points are relevant, clear, and interesting. I like the way you wove your results in with ecological and agricultural events of the past several decades: drought, conversion of pastureland to cropland, etc. NOTE: you have a key typo in line 422: were instead of where (at least I think that's what you meant). It would help to simplify that sentence altogether- as I would recommend throughout your manuscript.

Omit the paragraph on global economic factors. Your article addresses genetics of Brafords; focus on that. This article is much, much too long, and inclusion of tangential topics is one reason why it is so long.

These trends continue throughout the Discussion. While individual explanatory points are very good, this section includes too much repetition of patterns in the Results, too much inclusion of tangential information (e.g., trends in other breeds of cattle, history of Brafords and other breeds, economics...), and consistently wordy, unnecessarily complex sentences. Say what you want to say as simply and concisely as you can. Edit your sentences repeatedly until they are as tight and clear as possible.

FINAL WORD

What you have here is two or three articles, once addressing the history of Braford cattle, including its two sites of genesis (U.S. and Australia), its dispersal to South America and elsewhere in the world, and related trends in numbers, sizes and diversity of herds. Another is the focus of this article: a very detailed analysis of the genetic diversity and other patterns of population genetics in Australian-originated Braford cattle in four South American nations. Your data are more than sufficient to produce an important, interesting, valuable contribution to agricultural genetics. A third could be a socioeconomic-biogeographic study of trends in cattle herds (particularly the hybrid breeds like Brafords) across the globe. Your discussion includes a very nice start to such an analysis.

BUT including all these subtopics in this article make it much too long, too complex, too confusing... and overwhelming. Without getting into any details of your Discussion, no mention of your Conclusions, and an insufficient evaluation of your Results, my review is still six pages long. Many scientific articles are only a little bit longer than that! This is not a review paper, it's the report of an interesting and important empirical study. Focus on THAT in this article, tease out the other information, and write another manuscript (or two... or three). You have enough information to do that, and each of those big ideas deserves its own presentation without being tangled with your excellent, thorough pop gen study.

Thank you for your hard work- it's appreciated, and quite worthy of publication. Just, please, keep it focused so busy scholars and practitioners can digest it!

Author Response

REVIEWER 1

Word choice, grammar, sentence structure and general use of English. Overall, the quality of writing and use of English in this manuscript is excellent and only minimal editing is needed. Here are a few suggestions to improve the clarity and flow of your manuscript:

Word choice or omission. Occasionally, you select English words that are confusing within the context, or omit a key word, thereby changing the meaning of your sentence.

Examples from your simple summary: "...genetic diversity was conducted and shaped over the years." I think what you mean is: selective breeding was conducted, thereby shaping genetic diversity within South American Braford cattle. Selective breeding can influence genetic diversity, but it can't conduct it. Similarly, it would be less confusing in the last sentence of the abstract to say, "...the parameters of genetic diversity to suggest effective strategies for Braford breeders."

Response: We thank the reviewer for the time and effort made to review our manuscript. We applied suggestions as we truly think they improve its quality and readability.

Example from your Discussion (line 524-525). "Planification" is not a word. Try a simpler approach, "This is favorable, and probably due to better planning of matings...". Shorter, clearer, tighter writing.

Response: We agree and followed reviewer’s suggestion.

Sentence structure/order of clauses. Sometimes your sentence structure, particular word order, obscures your meaning or makes the sentence ambiguous. The first sentence of your abstract is a good example. I think you mean, "This study analyzes...of Braford cattle in South America from 1949-2019 to suggest effective strategies for breeding in the future." Clearly you're not suggesting breeding strategies to use in the past, but in the future, yet your current sentence is structured as if you are making recommendations for the past.

Response: We agree and followed reviewer’s suggestion.

Omitted words. Example from your abstract, "Even if adequate levels of diversity can be found, practices that consider the overuse of bulls..." Obviously a bull (or at least his semen) must be involved in any breeding effort; what you really mean is overuse of individual bulls, which clearly can lead to increases in F.

I realize these comments a very picky, but they are truly meant to help. Precise communication is critical in population genetics- even more so than in many other science disciplines, especially in an important applied study like yours!

Response: We totally agree and followed reviewer’s suggestion.

Introduction: the first paragraph is confusing. See above comments and apply to this paragraph. For example, you mean historical records, not theory. Partin & Hudgins breeding vs. Alto Adams Jr.???

Response: We rewrote this paragraph to improve clarity.

Evolution, adaptability, environment, and artificial selection: Use these terms with care! In the context of evolution, adaptation is a change in genetic structure or allele diversity in a population. The ability of a living individual to tolerate or adjust to conditions its environment is not adaptation in this context. When you mean the living animals fail to thrive (i.e., slow growth rate, low productivity, high parasite load), you can say the breed is not adapted to its current environmental conditions OR that it is not genetically suitable to its current environmental conditions. An individual organism cannot adapt evolutionarily.

Response: We totally agree and followed reviewer’s suggestion. We rewrote this paragraph to improve clarity.

Evolution: do not use evolution to mean the history of something (e.g., history of cattle associations, breeding programs). Given your focus on genetic diversity and structure, use evolution ONLY to mean the change in allele frequencies within a population, and/or the genetic structure of the Braford metapopulation in South America.

Artificial selection: Unless herds are feral, with multiple cows and bulls roaming and mating freely without human influence, ALL domestic breeding events are artificial selection- particularly if straws of semen are used via AI. Use the phrase selective breeding instead.

Response: We changed it across the body text.

Do not say "natural environmental conditions" unless you're contrasting with unnatural conditions (e.g., irrigation, air conditioning). Just say environment (the place) or environmental conditions (temperature, humidity, solar radiation, etc.).

Response: We changed it across the body text.

Last sentence of first paragraph: I think you mean to say, "Brahman and Hereford crossbreeding occurred in an attempt to improve biological performance of cattle." The last clause is confusing. Do you mean the hybrid Braford cattle were not economically viable, or that the Hereford cattle were not??

Response: We totally agree and followed reviewer’s suggestion. We rewrote this paragraph to improve clarity.

Fig. 1: Major issues with your labels. (i) the desirable Braford trait is NOT climate adaptability, it's adaptation to tropical and subtropical environments., i.e., breeders sought cattle crossbreedings whose progeny would immediately perform well in Australia, SE U.S., etc. (ii) tick hypersensitivity- not sensibility. (iii) hooves always have some condition (excellent, poor, satisfactory). Be specific: which hoof ailments need to be reduced? (iv) very close! Use reduced incidence of eye cancer.

Response: We agree and followed reviewer’s suggestion.

Change the wording of these key phrases throughout the manuscript.

INTRODUCTION

While it's okay to present a concise history of the breed in Australia and its introduction to South America, the rest of the history of associations, specific breeders, etc. is superfluous and confusing. Remove the detailing of the history of Braford cattle in the United States. Given the distinct development of these two populations of Brafords (e.g., use of Brahman cows vs. bulls to develop the breed) and lack of U.S.-origin Brafords in South America, the U.S. breed is irrelevant to your study.

Response: We understand the reviewer’s point, indeed we did not go deep in the history of the US Population for the same reasons. However, we think that considering the fact that there is an almost simultaneous origin of North American and Australian population, it is important to highlight that, initially (given undocumented American contributions may have occurred along the course of the history of the breed), it was Australian cattle which gave way to South American population instead of American population despite the apparent location proximity. Still, we clarified this in the body text.

Restructure the introduction so that the advantages of introducing Brahman alleles to Hereford cattle to help combat eye cancer, biological performance in tropical/subtropical climates, etc., are addressed in more detail right away. These distinctions of Brahmans (and therefore, Brafords) are critical. Reduce the details about the history of study of Braford resistance to heat stress, eye cancer, ticks. Just outline the advantages of the breed, concisely, to show why they're important in South America.

THEN detail the history of the development of Australian Brafords and their introduction to South America. Delete the detailed history of the cattle associations (irrelevant), though include mention of the various studbooks and how they were structured in each association. Include background information about the genetic diversity of Braford cattle, i.e., what was known prior to your study.

Response: We rewrote the introduction and restructured following the reviewer suggestions. However, this is the only information that was available prior to our study. It took us a while to find accurate sources from were to access it. Indeed, when we mention the evolution of associations it is because these were the only protection structure that existed.

What do you mean by population effectives? If you mean that the breeding programs resulted in several distinct effective genetic populations, then clarify.

Response: No, we meant reproductively active individuals. We clarified.

THEN state your study objectives. This structure will reduce redundancy in your manuscript and improve organization and clarity.

Response: We followed the reviewer’s suggestion.

The paragraph composed of lines 158-170 does not make sense. Convoluted & confusing. Rewrite the entire paragraph, and reorganize the ideas/sentences for better clarity and flow.

Response: We agree. We rewrote the paragraph following the reviewer’s suggestion.

The paragraph composed of lines 172-178 also does not make sense. Was your only objective to develop a model? Didn't you also use it to analyze F (inbreeding), C (coancestry), relatedness coefficent, etc., for Brafords in South America? What do you mean by "analysis of pedigree completeness downwards?" Define precisely "the population" you are studying (line 174 and elsewhere). The last sentence is a long, confusing run-on sentence. Divide into 2-3 simpler, clearer sentences. That sentence brings up "market demands in geographically diverse frameworks." Your study is not investigating market demands and how they vary across nations or other geographic entities- just delete that. If you want to specify Braford TRAITS that you hope to identify a genetic basis for through your study, fine. Focus on the population genetics, the genetic structure and allelic diversity of S. American Braford cattle.

Response: We agree. We rewrote the paragraph following the reviewer’s suggestion.

Given that you analyzed data only from the Asociasiόn Braford Argentina, clarify in your title and Introduction which population you investigated. Were all these cattle bred in Argentina? Table 1 indicates cattle from four countries were included in your study. Which nations does the ABA studbook include? Does it include ALL Brafords in all four nations, only cattle bred in Argentina, or ?? The Brafords across South America may form a distinct metapopulation, or the ASA cattle may form their own distinct metapopulation, restricted geographically to Argentina and/or the areas adjacent. These details are CRITICAL to include. [well... I finally found the information... in lines 313-314, in the Results. This information is fundamental for the METHODS, so include it there!]

Response: We followed the reviewer suggestion.

METHODS

Who is your intended audience? If you are only targeting population geneticists, then your shorthand use of pop gen jargon and statistical metrics is acceptable. If you are also targeting a more general audience composed of cattle breeders, animal scientists, agricultural educators and researchers, then you must briefly define what an F-statistic is (etc.). Pop gen is a jargon-filled discipline, and many scientists- never mind ranchers/breeders, won't know a lot of those details.

Response: We further clarified this for a broader audience as suggested.

If you analyzed studbook data to create Table 1, it should be placed in Results, and the techniques used to generate the summary metrics addressed in Methods- even though they are routine demographic analyses. It's unclear whether the ABA provided the metrics in Table 1, or whether you calculated or determined these details from information in the ABA studbook.

Response: Table 1 is correctly placed as data was extracted after the visual evaluation and exploration of data without having run any analysis yet.

Your control procedures, using only fully known bull-cow breedings, are good, but are presented in a confusing way. Clarify and explain why your control calculations are a good comparison to better reveal the implications of your analyses using the whole data set (entire ABA studbook).

Response: We clarified this in the body text.

Your analysis of ancestral contributions and gene origins is excellent... but poorly explained. How did you determine which bulls (and cows? Or no cows included?) were classified as founders?

Response: We clarified this in the body text.

Again, if your intended audience includes cattle breeders (or agricultural scientists who are not grounded in evolutionary biology and population genetics), you need to define and explain terminology like effective population sizeunequal (genetic) contributionsheterozygositygenetic drift, and genetic distance. If you expect only evolutionary biologists and population geneticists to read your article, then you need not define the many terms you include. Almost all of my undergraduate biology majors, and most of our masters-level graduate students, would be lost and confused- and they all study ecology, evolution and genetics. But population genetics and all the statistical procedures used to wring out information from large genetic data sets is rather specialized information; just a little bit of explanatory information/definitions would allow a huge potential audience to benefit from your work!

Response: We added definitions as suggested by the reviewer. However, these are basic concepts that can be found in any text book dealing with the matter.

RESULTS

Tables 3 and 4 are very cool! I found your results fascinating... and useful for breeders who are knowledgeable in population genetics. These tables ought to be in your Results section, however. This is a general weakness of your manuscript: placing information in the correct section. Some of your Introduction is better placed in your Methods; some of your Methods belong in the Introduction or Results; some of your Results belongs in the Methods.

Response: We changed the position of Tables and revised the whole manuscript for redistribution.

Please reorganize your article to put information where readers expect (and need) it to be.

The actual results are eye-opening and useful. Fewer, smaller herds? Not what I would have expected. Relatively more bulls contributing genetically, but fewer cows? Also unexpected- especially when the mean number of calves/cow increased so strongly. You put these outcomes clearly in context by investigating selectivity of bulls and cows, both of which increased. Figures 2 & 3 are very nice, but would benefit from a bit more information in the caption (or body of the Methods or Results) explaining what the Genetic Conservation IndexCoancestry (etc.) actually tell us about this population. Again, if you're targeting only pop gen experts, never mind.

Response: We added definitions as suggested by the reviewer. However, these are basic concepts that can be found in any text book dealing with the matter. The manuscript is already long, and adding basic definitions, unnecessarily extends it from our point of view. We must not forget that this is a paper not a textbook.

The paragraph beginning line 366 is much too long and dense; break it up into 2-3 shorter, more focused paragraphs. These results are complex and need to be digested in smaller chunks.

The fact that NONE of the individual farms, provinces or nations can be considered a genetic nucleus to these Braford cattle seems a critically important point, but it's deeply buried in the above, long paragraph (lines 382-384).

Wait... I'm confused. Your genetic distances analyses include populations from Australia? That wasn't clear before. Be sure each of the inclusions and exclusions from each of your analyses is clearly stated in the Methods, and not deeply buried. Your genetic distances outcomes are fascinating, however! I very much like Figures 6, 7 and 8. It was fun to peruse them in detail, and they support your narrative very well. It would be nice to have the individual farm, its location, and and province locations provided an Appendix listing the ID used in the genetic-distances figures. for each farm and province so readers can figure out which comparison each branch of the dendrogram represents. It's obvious for the nations, but not for the individual farms, their geographic locations, or those of the provinces.

Response: We followed the reviewer suggestions. We understand the reviewer proposal, but this cannot be done due to international data protection policies.

DISCUSSION

Overall, I like your discussion. The points are relevant, clear, and interesting. I like the way you wove your results in with ecological and agricultural events of the past several decades: drought, conversion of pastureland to cropland, etc. NOTE: you have a key typo in line 422: were instead of where (at least I think that's what you meant). It would help to simplify that sentence altogether- as I would recommend throughout your manuscript.

Omit the paragraph on global economic factors. Your article addresses genetics of Brafords; focus on that. This article is much, much too long, and inclusion of tangential topics is one reason why it is so long.

These trends continue throughout the Discussion. While individual explanatory points are very good, this section includes too much repetition of patterns in the Results, too much inclusion of tangential information (e.g., trends in other breeds of cattle, history of Brafords and other breeds, economics...), and consistently wordy, unnecessarily complex sentences. Say what you want to say as simply and concisely as you can. Edit your sentences repeatedly until they are as tight and clear as possible.

Response: We followed the reviewer suggestions. We did our best to improve the manuscript readability.

FINAL WORD

What you have here is two or three articles, once addressing the history of Braford cattle, including its two sites of genesis (U.S. and Australia), its dispersal to South America and elsewhere in the world, and related trends in numbers, sizes and diversity of herds. Another is the focus of this article: a very detailed analysis of the genetic diversity and other patterns of population genetics in Australian-originated Braford cattle in four South American nations. Your data are more than sufficient to produce an important, interesting, valuable contribution to agricultural genetics. A third could be a socioeconomic-biogeographic study of trends in cattle herds (particularly the hybrid breeds like Brafords) across the globe. Your discussion includes a very nice start to such an analysis.

BUT including all these subtopics in this article make it much too long, too complex, too confusing... and overwhelming. Without getting into any details of your Discussion, no mention of your Conclusions, and an insufficient evaluation of your Results, my review is still six pages long. Many scientific articles are only a little bit longer than that! This is not a review paper, it's the report of an interesting and important empirical study. Focus on THAT in this article, tease out the other information, and write another manuscript (or two... or three). You have enough information to do that, and each of those big ideas deserves its own presentation without being tangled with your excellent, thorough pop gen study.

Thank you for your hard work- it's appreciated, and quite worthy of publication. Just, please, keep it focused so busy scholars and practitioners can digest it!

Response: We thank the reviewer for his/her suggestions and thorough work on our manuscript. We approached each and every comment and did our best to improve readability. The information that we have here is indeed a third of the information that we obtained trough the evaluation of pedigree information (two additional papers are being prepared. However, and even if we understand the reviewer’s proposal, subdividing the information more would make it compulsory to perform additional methods to provide each of the manuscripts with sufficient entity. Furthermore, splitting the information here, may make results, which can be hard to understand if you are not acquainted with the topic, definitely hard to seek or poorly supported as it normally happens in literature, as very specific parts of the analysis of pedigree are approached and unconnectedly reported.

Reviewer 2 Report

The article is interesting since it allows to know the historical genetic evolution of the Bradford breed; however, some doubts and suggestions are detailed below.

Line 158. The objective set is quite ambitious. It is considered that the final part of the same "to suggest effective breeding strategies for breeds in need to fulfill market demands in geographically diverse frameworks" in the current state of the manuscript is scarcely elaborated.

Line 186. How do to explain the significant difference between the number of males and females in the historical population dataset? Is it due to the use of bulls over females of other races? Please explain. This could also explain the current low number of calves per mated bull and the decline in female selection (if only Bradford calves are considered)

In table 1, what is the reason for the current low number of calves per bull and the mean number of calves per mated bull, considering that the increase in the Mated Cow / Bull Ratio would expect an increase in the variable?

In line 198, the reference population is mentioned; it is suggested to include the information for this population in tables 1 and 2. Are the indicators annual or in the life of the animal?

Line 214. There are two "were calculated"
Line 222. It is pointed out that reference datasets are shown in table 2, which is incorrect.

Table 5. Consider a fixed number of decimals.
Line 298. The number of the table is lost.
Table 6. How are the negative inbreeding values in the subpopulation and total population explained?

Data and tables corresponding to results are presented in materials and methods.

Line 310-334. Although the historical and current population data are mentioned, the reference population is not mentioned.

Line 319. Which is the reason for the increase in bulls and decrease in cows?

Line 325. Average number of calves per mated bull decreased from 55.56 to 12.44, from the historical to the current population, while the average number of calves per mated cow increased from 1.75 to 6.39. Is this not contradictory with the greater number of females per male?

Line 363 and 364, eliminate two decimals

Figure 2. Although an explanation is included in the discussion, in relation to the peak of the year 1977 in birth number, this figure seems exaggerated. Please include an explanation of the phenomenon.

Line 438. Is the trend of a greater number of bulls than cows associated with reproductive technologies such as embryo transfers ?, since otherwise, it is not logical that there are fewer females of the race than males.

Line 511. On line 433, it is indicated that in 1977 the introduction of the breed in Latin American countries took place, but here it is indicated that it was in 1967.
Line 543. In this sentence, it is not understood if it is referring to the use of the original races (Braman and / or Angus) or to Braford itself.

Line 551. It is mentioned that ΔR was kept practically constant through the years; however, in addition to the peak mentioned for the year 2017, substantial decreases are observed in the years 1979 and 1989 and another peak in 1992

Line 588, What are the reasons for the respective bottlenecks that explain the low number of the effective number of ancestors (fa).
Line 591, in 450,154 replace "," with "."
Line 573-595. In the discussion, the values ​​obtained are compared with other publications; however, their implication is not discussed.

Line 596. The authors point out low fa / fe ratios in the population; it is understood that these would be caused by the low fa produced by bottlenecks, however, obtaining fa / fe less than 1 is not common. What could other factors be responsible?

Line 615 to 619 is only to repeat the results obtained; it is suggested to comment.

Line 652, If bottlenecks only explain 0.12% of the loss of genetic diversity, why is this factor so influential in estimating fa?

Line 698. It is pointed out that the results obtained may be the basis "to tailor specific strategies to be implemented internationally to ensure the preservation of genetic diversity in Braford cattle", however, no concrete proposal is made, which was one of the objectives set.

Author Response

REVIEWER 2

The article is interesting since it allows to know the historical genetic evolution of the Bradford breed; however, some doubts and suggestions are detailed below.

Line 158. The objective set is quite ambitious. It is considered that the final part of the same "to suggest effective breeding strategies for breeds in need to fulfill market demands in geographically diverse frameworks" in the current state of the manuscript is scarcely elaborated.

Response: Objectives were rewritten.

Line 186. How do to explain the significant difference between the number of males and females in the historical population dataset? Is it due to the use of bulls over females of other races? Please explain. This could also explain the current low number of calves per mated bull and the decline in female selection (if only Bradford calves are considered).

In table 1, what is the reason for the current low number of calves per bull and the mean number of calves per mated bull, considering that the increase in the Mated Cow / Bull Ratio would expect an increase in the variable?

Response: The number of calves per mated bull was reduced from the historical population to the current population, while the opposite occurred in the case of females, in which the number of calves per cow increased. These results may derive from the policies which indeed gave way to the Australian Braford ‘line’ and which may have continued in time and transferred to South American populations during the process of internationalization. Brahman was regarded as the breed to crossbred to Hereford in or-der for the later to achieve features at which it poorly performed. In the Australian ‘line’, Hereford cows participated of the cross contrary to what happened in the (North) American ‘line’. These results denote the appreciation of females which is also reflected on the historical number of females doubling the number of males. After-wards, the increasing implementation of reproductive biotechnologies used by producers, such as the use of embryo transfer, may have translated into an increase in the number of offspring per cow [59].

In line 198, the reference population is mentioned; it is suggested to include the information for this population in tables 1 and 2. Are the indicators annual or in the life of the animal?

Response: ENDOG (v4.9) software [28] takes as reference population those animals in the pedigree with both parents known, to compute fg as default. This was not considered an actual dataset but an intermedium step for certain calculations.

Line 214. There are two "were calculated"

Response: We removed it.

Line 222. It is pointed out that reference datasets are shown in table 2, which is incorrect.

Response: Any set under study is a reference dataset. Anyway, we removed it to clarify.

Table 5. Consider a fixed number of decimals.

Response: We corrected and used two decimals in the Table.

Line 298. The number of the table is lost.

Response: We corrected it.

Table 6. How are the negative inbreeding values in the subpopulation and total population explained?

Response: This indicated a small surplus of heterozygotes within herds.

Data and tables corresponding to results are presented in materials and methods.

Response: We moved Tables to the corresponding section.

Line 310-334. Although the historical and current population data are mentioned, the reference population is not mentioned.

Response: Any set under study is a reference dataset. This was not considered an actual dataset but an intermedium step for certain calculations. We understand the misconception and we corrected it.

Line 319. Which is the reason for the increase in bulls and decrease in cows?

Response: This finding may be supported on the fact that the Braford breed is mainly derived for meat production, hence, the interest for bulls is predominant against the interest in cows.  Such an interest was the causative agent of the increase in the proportion of males in the population along history, as a consequence of the increase in the reposition of bulls and to the commercialization of cows, as it was reported for most of the properties.

Line 325. Average number of calves per mated bull decreased from 55.56 to 12.44, from the historical to the current population, while the average number of calves per mated cow increased from 1.75 to 6.39. Is this not contradictory with the greater number of females per male?

Response: Not in the context of embryo transfer as pointed out in the body text.

Line 363 and 364, eliminate two decimals

Response: For these decimals it is necessary to use more than 2 given the range of values is lower and small increases are significant.

Figure 2. Although an explanation is included in the discussion, in relation to the peak of the year 1977 in birth number, this figure seems exaggerated. Please include an explanation of the phenomenon.

Response: An explanation was provided.

Line 438. Is the trend of a greater number of bulls than cows associated with reproductive technologies such as embryo transfers ?, since otherwise, it is not logical that there are fewer females of the race than males.

Response: Yes, indeed, we clarified.

Line 511. On line 433, it is indicated that in 1977 the introduction of the breed in Latin American countries took place, but here it is indicated that it was in 1967.

Response: We corrected it.

Line 543. In this sentence, it is not understood if it is referring to the use of the original races (Braman and / or Angus) or to Braford itself.

Response: We clarified it.

Line 551. It is mentioned that ΔR was kept practically constant through the years; however, in addition to the peak mentioned for the year 2017, substantial decreases are observed in the years 1979 and 1989 and another peak in 1992.

Response: These decreases in 1979 and 1989 or the increase in 1992 are not substantial in the context of the average of around 0.006%, which is what the body text states. The peak in 2017 is 4 times the values in the other peaks identified by the reviewer. We clarified this in the body text.

Line 588, What are the reasons for the respective bottlenecks that explain the low number of the effective number of ancestors (fa).

Response: In the present study, the estimated fa in the seven bovine populations analyzed was lower than their respective fe, which indicates a decrease in genetic diversity due to the presence of genetic bottlenecks in all population sets from their foundation. This imbalance in the values of fa and fe have been reported for breeds suffering a process of internationalization from a specific point of origin. From a genetic diversity perspective, the relocation or reintroduction of a section of a breed to a new place to serve as the founder basis of a new population, may produce a similar effect as a sharp reduction in the original population, given a relatively limited number of animals may serve as the population  basis on which the international expansion of such breed will be built [1]. Still, this effect is somehow buffered when the breed in particular is made out of a crossbreeding process between two or more breeds, as it occurs in Brafords. Furthermore, in the case of Brafords, this buffering effect is specially reinforced as in South America different percentages of blood of each of the participating breeds were tested and permitted in the standard seeking the particular economic interest of each country.

Such a buffering effect is denoted by the fact that, in the current study, the number of founder genome equivalents (fg), or the number of equally-contributing founders that would be expected to produce the same genetic diversity as observed in the current population if there is no random loss of founder alleles in descendants (e.g., through genetic drift) was as high as 420,154.

Line 591, in 450,154 replace "," with "."

Response: replaced.

Line 573-595. In the discussion, the values ​​obtained are compared with other publications; however, their implication is not discussed.

Response: We discussed the results as suggested.

Line 596. The authors point out low fa / fe ratios in the population; it is understood that these would be caused by the low fa produced by bottlenecks, however, obtaining fa / fe less than 1 is not common. What could other factors be responsible?

Response: We disagree, obtaining values of fa/fe lower than 1 are quite common in endangered, reintroduced, internationalized or fragmented populations. We explained this in the body text. We added citations to three papers to reinforce this statement.

Line 615 to 619 is only to repeat the results obtained; it is suggested to comment.

Response: We discussed the results as suggested.

Line 652, If bottlenecks only explain 0.12% of the loss of genetic diversity, why is this factor so influential in estimating fa?

Response: Rather than in fa, which is the minimum number of ancestors - which can be founders or not - needed to explain the genetic diversity of the current population, the number and severity of bottlenecks will be reflected in the difference between fa and fe. If there have been no population bottlenecks, fa will equal fe.

Line 698. It is pointed out that the results obtained may be the basis "to tailor specific strategies to be implemented internationally to ensure the preservation of genetic diversity in Braford cattle", however, no concrete proposal is made, which was one of the objectives set.

Response: We agree and added a concrete proposal as suggested.

Reviewer 3 Report

General comments

This study aimed to evaluate the Population Structure and Genetic Diversity Evolution of Bradfort breed un South America. The study is well conducted and very informative in discussion and conclusions.

Some few issues were detected: the simple summary is a background and not a summary. Please rewrite. Use the past instead of the present. Please round to one decimal in text for measurements. Figures 7 and 8 are results and should be reported in the proper section.

Specific comments

L53-54: Please re-write.

L63: “…(38.8 oC to 39.6 oC)…”; please round to one decimal in text for measurements.

L68-69: Can you add a reference?

L87-95: Can you add a reference? Mainly for L95.

L213-217: Please apply the norms of the journal for citations.

Author Response

REVIEWER 3

General comments

This study aimed to evaluate the Population Structure and Genetic Diversity Evolution of Bradfort breed un South America. The study is well conducted and very informative in discussion and conclusions.

Response: We thank the reviewer for his/her kind comments.

Some few issues were detected: the simple summary is a background and not a summary. Please rewrite. Use the past instead of the present. Please round to one decimal in text for measurements. Figures 7 and 8 are results and should be reported in the proper section.

 Response: Suggestions were followed.

Specific comments

L53-54: Please re-write.

Response: Rewritten.

L63: “…(38.8 oC to 39.6 oC)…”; please round to one decimal in text for measurements.

Response: We followed reviewer suggestion.

L68-69: Can you add a reference?

Response: Added.

L87-95: Can you add a reference? Mainly for L95.

Response: Added.

L213-217: Please apply the norms of the journal for citations.

Response: Applied.

Round 2

Reviewer 1 Report

This revised version of your excellent study is much clearer, better organized, and thereby more effective than the version first submitted. Thank you for your diligent work on this revision- it makes a very positive difference!

My comments to the editors follow:

The authors clearly worked diligently on their revision, infusing many of my recommendations and those of my fellow reviewers. While I still believe this contribution would be of superior clarity and usefulness if split into two, shorter articles (one addressing the history of the breed, its standards, studbooks and professional associations, and one the population genetics), it is cohesive- if quite long and overwhelmingly full of details- and reports an ambitious, well-designed study. I recommend its publication despite my personal preference for two shorter, more focused contributions vs. one large, comprehensive one. This is good work!